# Precisely timed dopamine signals establish distinct kinematic representations of skilled movements

Alexandra Bova[1], Matt Gaidica[1], Amy Hurst[2], Yoshiko Iwai[2], Julia Hunter[2], Daniel K Leventhal[2,3,4,5]*

[1]Neuroscience Graduate Program, University of Michigan, Ann Arbor, United States; [2]Department of Neurology, University of Michigan, Ann Arbor, United States; [3]Department of Biomedical Engineering, University of Michigan, Ann Arbor, United States; [4]Parkinson Disease Foundation Research Center of Excellence, University of Michigan, Ann Arbor, United States; [5]Department of Neurology, VA Ann Arbor Health System, Ann Arbor, United States

**Abstract** Brain dopamine is critical for normal motor control, as evidenced by its importance in Parkinson Disease and related disorders. Current hypotheses are that dopamine influences motor control by 'invigorating' movements and regulating motor learning. Most evidence for these aspects of dopamine function comes from simple tasks (e.g. lever pressing). Therefore, the influence of dopamine on motor skills requiring multi-joint coordination is unknown. To determine the effects of precisely timed dopamine manipulations on the performance of a complex, finely coordinated dexterous skill, we optogenetically stimulated or inhibited midbrain dopamine neurons as rats performed a skilled reaching task. We found that reach kinematics and coordination between gross and fine movements progressively changed with repeated manipulations. However, once established, rats transitioned abruptly between aberrant and baseline reach kinematics in a dopamine-dependent manner. These results suggest that precisely timed dopamine signals have immediate and long-term influences on motor skill performance, distinct from simply 'invigorating' movement.

*For correspondence:
dleventh@med.umich.edu

## Introduction

Brain dopamine plays a critical role in motor control. This is most clearly exemplified by the motor symptoms of Parkinson Disease (PD), in which brain dopamine levels are reduced. PD is defined by tremor, rigidity, bradykinesia, and postural instability. Bradykinesia and rigidity consistently improve with dopamine replacement, although tremor and postural instability may not. PD patients also experience significant disability from impaired manual dexterity, which causes difficulty with tasks like tying shoelaces, fastening buttons, and handwriting (*Pohar and Allyson Jones, 2009*). This symptom is distinct from bradykinesia (*Foki et al., 2016*), but also responds to dopamine replacement (*Gebhardt et al., 2008*; *Lee et al., 2018*). Thus, dopamine plays an important, but poorly defined, role in dexterous skill beyond simply regulating movement speed or amplitude.

Two leading hypotheses regarding the role of dopamine in motor control are that it 'invigorates' movement and regulates motor learning. The 'vigor' hypothesis derives from the exquisite dopa-responsiveness of bradykinesia in PD and is supported by extensive experimental evidence. Intra-striatal infusion of dopamine agonists increases locomotion, and both electrical and optogenetic stimulation of midbrain dopamine neurons cause contraversive turning (*Arbuthnott and Unger-stedt, 1975*; *Saunders et al., 2018*). Dopamine signaling increases near movement onset and acceleration bouts (*Coddington and Dudman, 2018*; *Howe and Dombeck, 2016*; *Jin and Costa, 2010*;

Schultz et al., 1983) and is correlated with movement velocity (*Barter et al., 2015*; *Saunders et al., 2018*). Conversely, dopamine depletion and dopamine receptor blockade slow movement (*Leventhal et al., 2014*; *Panigrahi et al., 2015*). These studies used scalar readouts that reflect 'vigor' (e.g. movement velocity or numbers of rotations), and therefore could not assess dopaminergic influences on multi-joint coordination.

Dopaminergic roles in reinforcement learning may contribute to 'non-vigor' aspects of motor control. Phasic dopamine release patterns are broadly consistent with 'reward prediction error' (RPE) signals, or the difference in value between anticipated and realized behavioral states (*Glimcher, 2011*). In reinforcement learning models, the RPE is used to adjust subsequent behavior. While the details of dopamine's role in implicit learning remain to be fully elucidated (*Schultz, 2019*), dopamine signaling clearly influences synaptic plasticity and alters future behavior (*Dowd and Dunnett, 2005*; *Leventhal et al., 2014*; *Mohebi et al., 2019*; *Parker et al., 2016*; *Shen et al., 2008*). Most evidence for 'learning' models of dopamine function come from behavioral tasks that require no movement (e.g. classical conditioning, *Tobler et al., 2005*), simple movements (e.g. lever presses, *Parker et al., 2016*), or innate movements (e.g. locomotion, *Howe and Dombeck, 2016*). For the most part, such tasks have discrete outcomes (e.g. push the right or left lever, initiate locomotion or not). However, dopaminergic roles in instrumental and classical conditioning may extend to tasks with more degrees of freedom. In support of this hypothesis, dopamine neuron firing patterns consistent with RPEs (more accurately, performance prediction errors) are observed in songbirds receiving distorted audio feedback (*Gadagkar et al., 2016*). In mice, rotarod performance worsens gradually during dopamine receptor blockade, and improves gradually when the blockade is released (*Beeler et al., 2012*). These results could be explained by dopamine reinforcing specific, successful actions (e.g. paw adjustments on the rotarod) to gradually improve performance (*Beeler et al., 2013*). Nonetheless, the role of dopamine in skilled, dexterous movements requiring precise multi-joint coordination remains unclear.

The goal of this study was to determine the effects of precisely timed dopaminergic manipulations on a complex, finely coordinated, and relatively unconstrained motor skill. To do this, we optogenetically stimulated or inhibited midbrain dopamine neurons as rats performed a skilled reaching task. In skilled reaching, rats learn the coordinated forelimb and digit movements to reach for, grasp, and consume sugar pellets. Skilled reaching is readily learned by rats over several sessions (*Klein et al., 2012*; *Lemke et al., 2019*), requires precise coordination between the forelimb and digits, and is sensitive to dopamine depletion (*Hyland et al., 2019*; *Whishaw et al., 1986*). It is therefore an excellent model for assessing dopaminergic contributions to dexterous skill.

By combining skilled reaching, optogenetics, and measurement of three-dimensional paw/digit kinematics, we addressed the following questions. First, we asked whether dopamine manipulations affect current or subsequent reaches. If dopamine affects only the current movement, reach kinematics should change immediately with dopamine manipulations. Conversely, if dopamine provides a teaching signal for fine motor coordination, reach kinematics should depend on the history of prior dopaminergic activation. Second, we asked how reach kinematics – specifically coordination between forelimb and digit movements – are influenced by dopamine manipulations. If dopamine plays a purely 'invigorating' role in movement, altered dopaminergic signaling should affect only the velocity or amplitude of the reaches.

Instead of pure vigor or learning roles for dopamine, we found a complex pattern of dopaminergic influences on skilled reaching. Consistent with a motor learning function, reach kinematics changed gradually with repeated dopamine neuron stimulation or inhibition. In addition to simple kinematic measures (e.g. reach amplitude), coordination between paw advancement and digit movements also changed with repeated stimulation/inhibition. However, once established, rats transitioned between aberrant and baseline reach kinematics within a single trial in a dopamine-dependent manner. These results indicate that dopamine has both immediate and long-term effects on motor control beyond simply invigorating movement, with important implications for understanding dopamine-linked movement disorders.

## Results

We optogenetically stimulated or inhibited substantia nigra pars compacta (SNc) dopamine neurons at specific moments during rat skilled reaching. Tyrosine hydroxylase (TH)-Cre[+] rats were injected

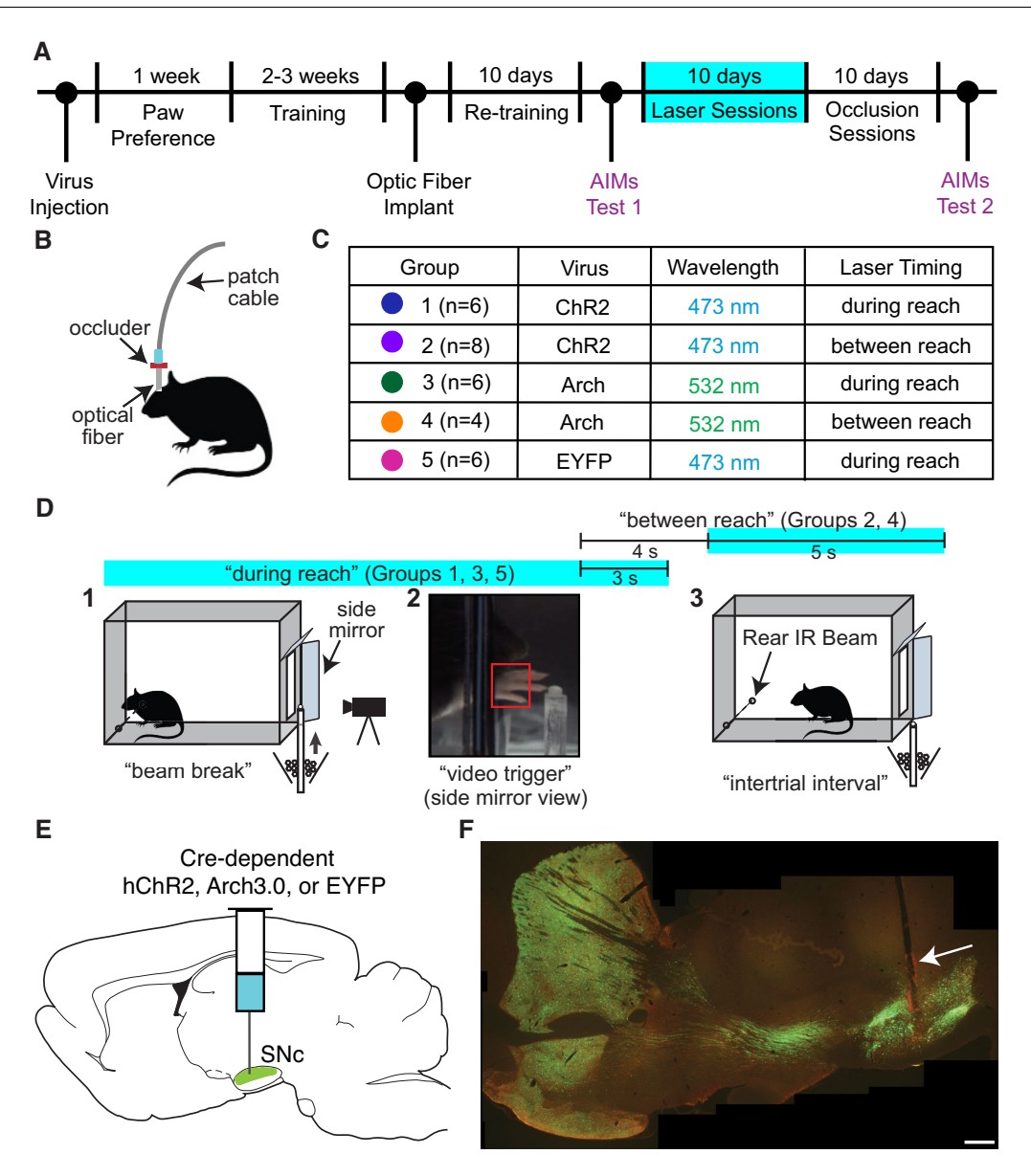

**Figure 1.** Experimental framework. (A) Timeline for a single experiment. AIMs Test – Abnormal Involuntary Movement testing (see 'Dopamine neuron stimulation induces context- and history-dependent abnormal involuntary movements'). The 'ChR2 between' group also received 'during reach' stimulation after 'occlusion' sessions (see *Figure 11*). (B) Light was physically occluded from entering the brain by obstructing the connection between the optical fiber and patch cable during 'occlusion' sessions. (C) Rats were assigned to one of five groups based on virus injected and timing of optogenetic manipulation. *n* is the number of rats included in the analysis for each group (see Materials and methods). Dot colors correspond with the color used to represent each group in subsequent figures. (D) A single skilled reaching trial. 1 – rat breaks IR beam at the back of the chamber to request a sugar pellet ('beam break'). 2 – Real-time analysis detects the paw breaching the reaching slot to trigger 300 fps video from 1 s before to 3.33 s after the trigger event ('video trigger'). 3 – 2 s after the trigger event, the pellet delivery rod resets and the rat can initiate a new trial ('intertrial interval'). Optogenetic manipulations occurred either during reaching (beam break to 3 s after 'video trigger') or between reaches (beginning 4 s after 'video trigger' and lasting 5 s). *Figure 1—figure supplement 1* shows the distribution of the duration of 'during reach' laser-on epochs. (E) Double-floxed ChR2-EYFP, Arch-EYFP, or control EYFP constructs were injected bilaterally into SNc. (F) Immunohistochemistry against EYFP showing expression of a fused ChR2-EYFP construct in the nigrostriatal pathway. Optical fibers (arrow) were implanted over SNc contralateral to the rat's preferred reaching paw. Estimated locations of all fiber tips are shown in *Figure 1—figure supplement 2*. Representative immunohistochemistry images from each group are shown in *Figure 1—figure supplement 3*. Scale bar = 1 mm.

The online version of this article includes the following source data and figure supplement(s) for figure 1:

**Figure supplement 1.** The duration of 'during reach' and 'between reach' stimulation is closely matched.

*Figure 1 continued on next page*

*Figure 1 continued*

**Figure supplement 1—source data 1.** A .mat file containing the durations of each 'laser on' bout in the 10 'laser on' sessions for each rat from 'ChR2 During', 'ChR2 Between', 'Arch During', and 'Arch Between' groups.

**Figure supplement 2.** Optical fiber locations.

**Figure supplement 2—source data 1.** A .xlsx file containing the estimated fiber tip locations for each rat in all groups.

**Figure supplement 3.** Representative immunohistochemistry against EYFP showing expression of fused opsin-EYFP constructs in the nigrostriatal pathway for all groups.

bilaterally with a double-floxed channelrhodopsin (ChR2), archaerhodopsin (Arch), or control EYFP construct into SNc (*Figures 1A, C and E*). Rats were trained on an automated skilled reaching task that allows synchronization of high-speed video with optogenetics (*Bova et al., 2019*; *Ellens et al., 2016*). Trials started with rats breaking a photobeam at the back of the chamber, which caused a pellet to be delivered in front of the reaching slot (*Figure 1D*). Rats could make multiple reaches until the pellet delivery arm descended 2 s after the video trigger event. Following training, optical fibers were implanted over SNc contralateral to the rat's preferred reaching paw. Immunohistochemistry confirmed that opsin expression was restricted to TH-expressing neurons in SNc projecting to striatum (*Figure 1F* and *Figure 1—figure supplement 3*).

## Altered SNc dopamine neuron activity gradually changes skilled reaching outcomes

We stimulated or inhibited SNc dopamine neurons during every reach for ten 30-min sessions (*Figure 1D*, 'during reach'). Baseline performance did not differ between groups (*Figure 2*). Dopamine neuron stimulation did not significantly affect the number of trials attempted (*Figure 2B*), but caused a progressive decline in performance (*Figure 2E* and *Figure 2—figure supplement 2*). Success rate on the first reach within each trial decreased to about half of baseline performance during the first day of testing, then to nearly 0% for the remainder of 'Laser On' sessions. The number of reaches per trial increased as success rate declined, but even with multiple attempts per trial, success rate decreased (*Figure 2—figure supplement 2*). The progressive decline in performance led us to ask whether success rate also changed across trials within individual sessions. Indeed, during the first 'Laser On' session, success rate progressively declined (*Figure 2G*, *Figure 2—figure supplement 1*). Furthermore, dopamine-dependent changes in reach success persisted into subsequent sessions. Therefore, reaching performance depends on the history of dopamine neuron activation during skilled reaching.

Because dopamine stimulation during reaching caused a gradual decline in performance, we asked if reaching performance would recover gradually when dopamine stimulation was removed. Animals were tested for an additional 10 days with the same laser stimulation protocol, but with the patch cable-optical fiber junction physically occluded ('occlusion' sessions, *Figure 1B*). Thus, all cues were identical (e.g., optical shutter noise, visible light) except light penetration into the brain. Reaching performance recovered quickly, but not immediately, to pre-stimulation levels (*Figure 2E,G*). However, there was significant variability between rats in the rate of recovery (*Figure 2—figure supplement 1*). On average, recovery to baseline performance was faster than the decline in performance with initial dopamine stimulation (contrast testing, $t(583.8) = 2.55$, p=0.011). This is further evidence that the history of dopaminergic activation influences subsequent skill execution.

We next asked if dopamine stimulation must occur during reaches to affect success rate. A separate group of ChR2-expressing TH-Cre[+] rats received laser stimulation during the intertrial interval for a duration matched to 'during reach' stimulation (*Figure 1D*, 'between reach', *Figure 1—figure supplement 1*). Dopamine neuron stimulation between reaches did not significantly affect the number of trials (*Figure 3A*) or success rate (*Figure 3B,C* and *Figure 3—figure supplements 1,2*). Therefore, dopamine neuron stimulation must occur as the rat is reaching to affect subsequent reaching performance. This result has two important implications. First, it suggests that skill performance depends on the history of striatal dopamine levels specifically during performance of that skill. Second, it argues against the possibility that the effects of dopamine neuron stimulation are due to the gradual accumulation of striatal dopamine.

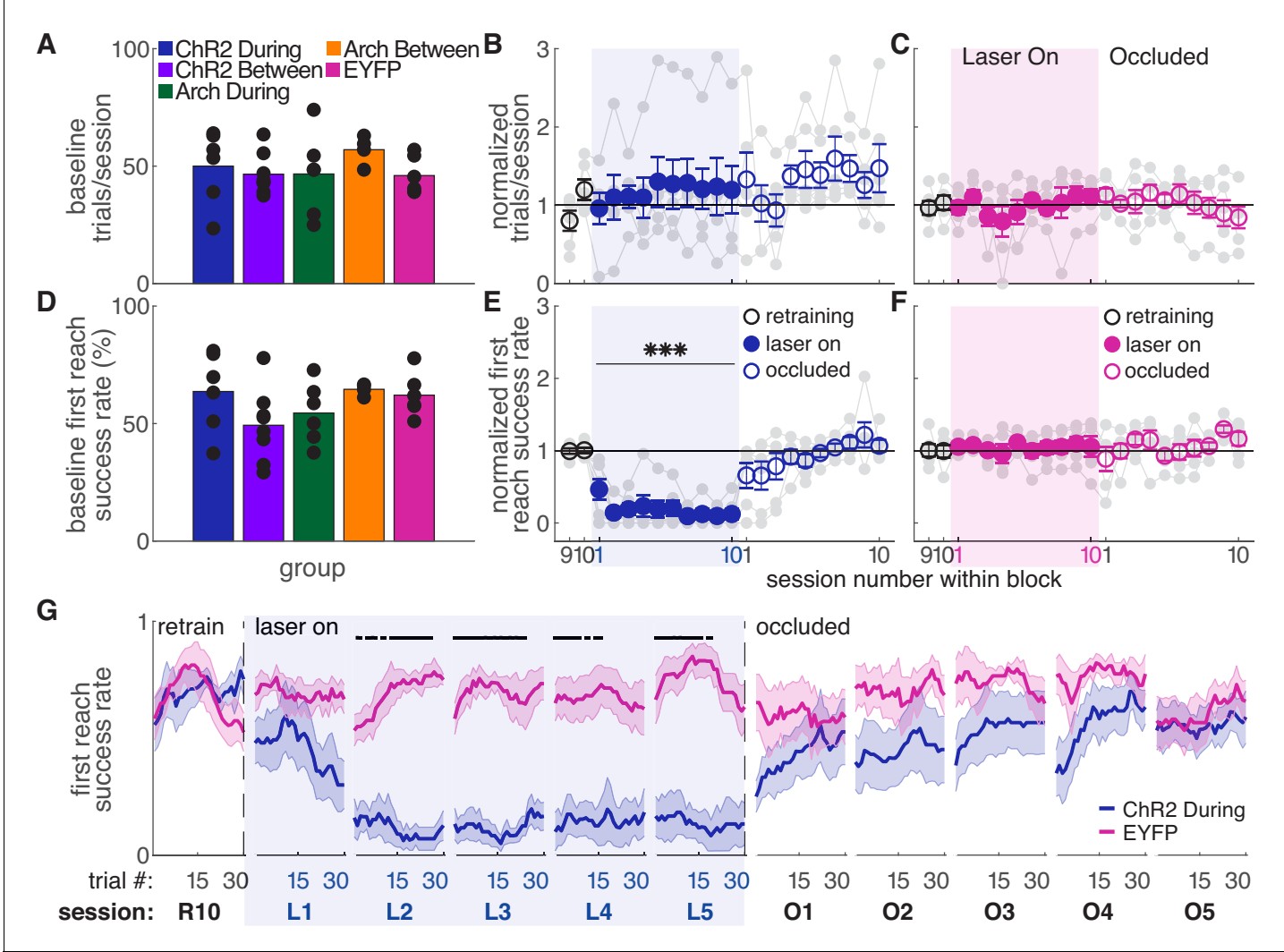

**Figure 2.** Dopamine neuron stimulation during reaches gradually impairs skilled reaching performance. (A) Average number of trials per session over last two 'retraining' sessions for each group. Black dots represent individual rats. Baseline number of reaches performed did not differ between groups. Kruskal-Wallis Test: $\chi^2(4)=3.94$, p=0.41. (B) Average number of trials per session divided by the baseline number of trials for 'during reach' stimulation. Gray lines represent individual rats. Linear mixed model: effect of laser: $t(79) = 0.932$, p=0.35; interaction between laser and session: $t(584) = -0.99$, p=0.32. (C) Same as (B) for control rats injected with an EYFP-only construct. Linear mixed model: effect of laser: $t(79) = -0.90$, p=0.37; interaction between laser and session: $t(584) = 1.20$, p=0.23. (D) Average first attempt success rate over the last two 'retraining' sessions for each group. Black dots represent individual rats. Baseline success rate did not differ between groups. Kruskal-Wallis Test: $\chi^2(4)=6.18$, p=0.19. (E) Average first attempt success rate divided by baseline success rate for 'during reach' stimulation. Linear mixed model: effect of laser: $t(133) = -3.76$, $p=2.51\times10^{-4}$; interaction between laser and session: $t(584) = -1.50$, p=0.13. (F) Same as (E) for control rats injected with an EYFP-only construct. Linear mixed model: effect of laser: $t(134) = 0.63$, p=0.53; interaction between laser and session: $t(584) = -0.42$, p=0.67. (G) Moving average of success rate within individual sessions in the last retraining session, first 5 'laser on' sessions, and first 5 'occlusion' sessions. Unlike panels E and F, these are not normalized to retraining sessions because they are within-session moving averages. Black bars represent statistically significant differences between groups (Wilcoxon rank sum test, p<0.01). 'R10', 'L1', 'O1', etc. indicate the 10th retraining session, first 'laser on' session, first 'occlusion' session, etc. Shaded colored areas in (G) and error bars in B-C and E-F represent standard errors of the mean (s.e.m). Data for individual rats are shown in *Figure 2—figure supplement 1*. *Figure 2—figure supplement 2* shows additional performance measures for 'during reach' ChR2 activation and EYFP control rats ('any reach' success rates, number of attempts per trial, and breakdown of reach outcomes across sessions). *** Indicates p<0.001 for the laser term in the linear mixed model in panel E.

The online version of this article includes the following source data and figure supplement(s) for figure 2:

**Source data 1.** A .mat file containing number of trials (num_trials) and first reach success rate (firstReachSuccess) for 22 testing sessions ('retraining', 'laser on', and 'occluded').

**Source data 2.** A .mat file containing first reach success rate averages across a moving block of 10 trials for 22 testing sessions ('retraining', 'laser on', and 'occluded').

*Figure 2 continued on next page*

*Figure 2 continued*

**Source data 3.** Statistics.
**Figure supplement 1.** Moving average of success rate within individual sessions (ChR2 During and EYFP) for each rat.
**Figure supplement 1—source data 1.** A .mat file containing first reach success rate averages across a moving block of 10 trials for 22 testing sessions ('retraining', 'laser on', and 'occluded').
**Figure supplement 2.** Additional task performance measures for 'during reach' stimulation and EYFP controls.
**Figure supplement 2—source data 1.** A .mat file containing any reach success rate (anyReachSuccess) and number of reach attempts (mean_-num_reaches) for 22 testing sessions ('retraining', 'laser on', and 'occluded').
**Figure supplement 2—source data 2.** A .mat file containing percentage of trials that were each outcome type for each session for each rat (fullOutcomePercent).
**Figure supplement 2—source data 3.** Statistics.

Dopamine neuron inhibition during reaching did not affect success rate (*Figure 4C,E* and *Figure 4—figure supplements 2,3*). However, dopamine neuron inhibition significantly decreased the number of trials per session (*Figure 4A*), consistent with a role for midbrain dopamine in motivation to work for rewards (*Palmiter, 2008*; *Salamone and Correa, 2012*). It is not clear why dopamine neuron stimulation did not have the opposite effect (*Figures 2B,3A*), but increases in trial numbers may be limited by a ceiling effect - rats initiate new trials quickly, even in the absence of stimulation. The inhibition-related decrease in trials was gradual, with rats progressively performing fewer reaches across sessions. Dopamine neuron inhibition between reaches had no effect on success rate (*Figure 4D* and *Figure 4—figure supplement 1*) or the number of reaches performed in each session (*Figure 4B*). Control rats injected with constructs expressing EYFP but no opsin did not experience any changes in task performance (*Figure 2C,F,G* and *Figure 2—figure supplements 1,2*).

## Dopamine manipulations induce progressive changes in reach-to-grasp kinematics

The success rate analysis indicates that repeated dopaminergic stimulation progressively diminished reaching performance, but does not explain why performance worsened. To determine which aspects of reach kinematics were altered by dopaminergic manipulations, we used Deeplabcut to track individual digits, the paw, and the pellet (*Figure 5*; *Bova et al., 2019*; *Mathis et al., 2018*).

Consistent with the success rate analysis, dopamine neuron stimulation during reaching caused progressive changes in reach-to-grasp kinematics. Reach extent (how far the paw extended in the direction of the pellet, $z_{digit2}$) became progressively shorter with repeated stimulation during reaches (*Figure 6A*, *Videos 1* and *2*). This effect may have been stronger for posteromedially located fibers (*Figure 6—figure supplement 1*). The progressive change in reach extent occurred both across and within sessions, and did not stabilize until the fifth session of dopamine neuron stimulation (*Figure 6D* and *Figure 6—figure supplement 3*). Dopamine neuron stimulation during reaches also gradually narrowed grasp aperture at reach end (*Figure 6E,F* and *Figure 6—figure supplement 4*), caused the paw to be more pronated at reach end (i.e. θ decreased) (*Figure 6G,H* and *Figure 6—figure supplement 5*), and decreased the maximum reach velocity (*Figure 6I,J* and *Figure 6—figure supplement 6*). Kinematic measures continued to change even when success rate had plateaued (compare *Figures 2E,6*) due to a 'floor effect' for success rate – once the rat consistently misses the pellet, no further changes are detectable by this measure. The decrease in reach extent accounted for much of the drop in success rate (*Figure 6—figure supplement 7*), although reaches matched for reach extent were still less successful during 'laser on' than 'occlusion' sessions (*Figure 6—figure supplement 8*). This suggests that paw transport and grasp-related factors both contributed to reach-to-grasp failures. When dopamine stimulation ceased ('occlusion' sessions), reach-to-grasp kinematics rapidly returned to baseline. As with success rate, there was individual variability in how quickly rats returned to pre-stimulation kinematics (*Figure 6A,D–J* and *Figure 6—figure supplements 3–6*). All reach-to-grasp kinematics were unchanged in rats receiving dopamine neuron stimulation between reaches and EYFP control rats (*Figure 6B–D,F,H,J* and *Figure 6—figure supplements 2–8*). In addition to histology, we verified opsin expression and fiber placement by performing 'during reach' stimulation in rats previously stimulated between reaches. All rats showed

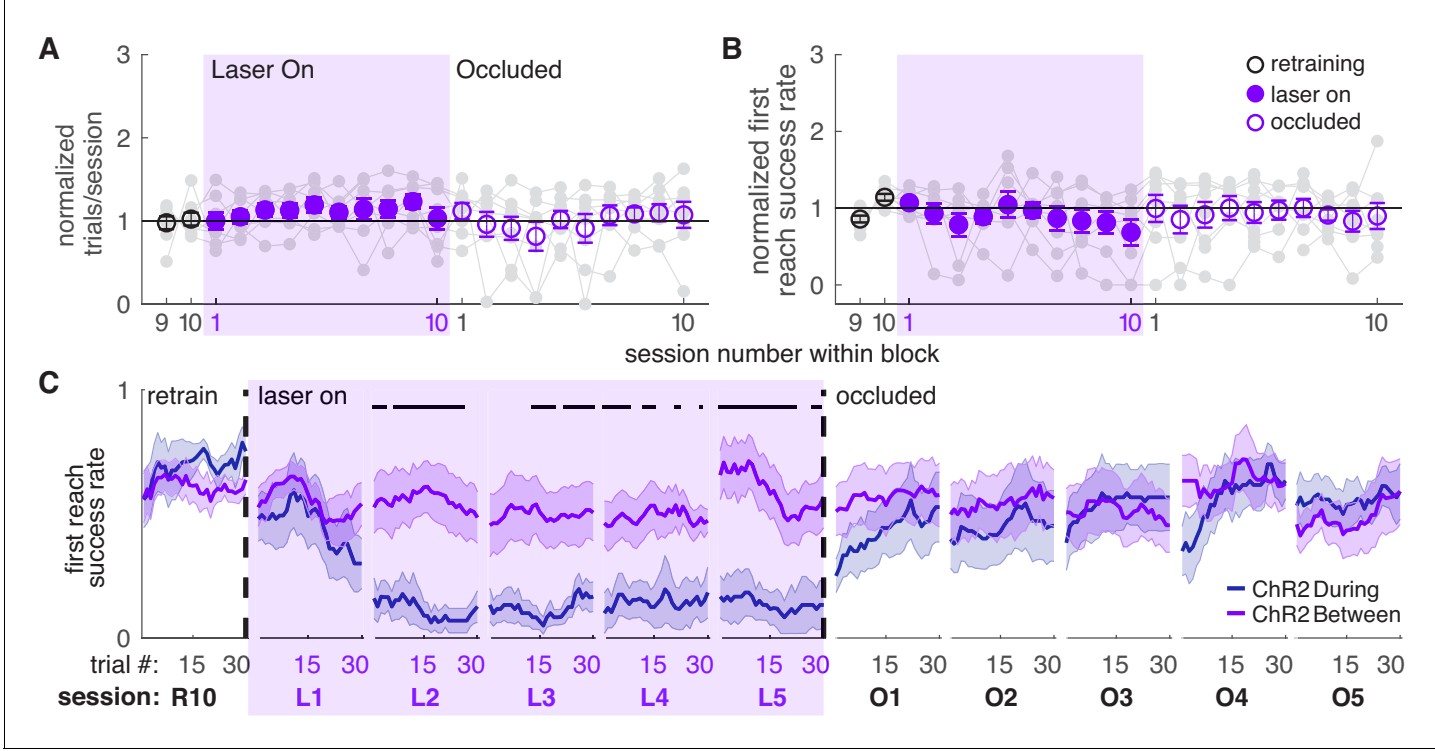

**Figure 3.** Dopamine neuron stimulation between reaches does not affect skilled reaching performance. (A) Average number of trials per session divided by the baseline number of trials for 'between reach' stimulation. Gray lines represent individual rats. Linear mixed model: effect of laser: t(79) = 1.13, p=0.26; interaction between laser and session: t(584) = −0.64, p=0.52. (B) Average first attempt success rate divided by baseline success rate for 'between reach' stimulation. Linear mixed model: effect of laser: t(133) = −0.29, p=0.78; interaction between laser and session: t(584) = −0.94, p=0.35. (C) Moving average of success rate within individual sessions in the last retraining session, first 5 'laser on' sessions, and first 5 'occlusion' sessions. 'R10', 'L1', 'O1', etc. indicate the 10th retraining session, first 'laser on' session, first 'occlusion' session, etc. 'During reach' data from *Figure 2* are shown for comparison. Unlike panel B, these are not normalized to retraining sessions because they are within-session moving averages. Black bars represent trials with a statistically significant difference between groups (Wilcoxon rank sum text, p<0.01). Data for individual rats are shown in *Figure 3—figure supplement 1*. *Figure 3—figure supplement 2* shows additional performance measures for 'between reach' ChR2 activation ('any reach' success rates, number of attempts per trial, and breakdown of reach outcomes across sessions). Shaded colored areas in C and error bars in A-B represent s.e.m.

The online version of this article includes the following source data and figure supplement(s) for figure 3:

**Source data 1.** A .mat file containing number of trials (num_trials) and first reach success rate (firstReachSuccess) for 22 testing sessions ('retraining', 'laser on', and 'occluded').
**Source data 2.** A .mat file containing first reach success rate averages across a moving block of 10 trials for 22 testing sessions ('retraining', 'laser on', and 'occluded').
**Source data 3.** Statistics.
**Figure supplement 1.** Individual rat data for moving average of success rate across trials within individual sessions ('ChR2 Between').
**Figure supplement 1—source data 1.** A .mat file containing first reach success rate averages across a moving block of 10 trials for 22 testing sessions ('retraining', 'laser on', and 'occluded').
**Figure supplement 2.** Additional task performance measures for 'between reach' ChR2 stimulation.
**Figure supplement 2—source data 1.** A .mat file containing any reach success rate (anyReachSuccess) and number of reach attempts (mean_num_reaches) for 22 testing sessions ('retraining', 'laser on', and 'occluded').
**Figure supplement 2—source data 2.** A .mat file containing percentage of trials that were each outcome type for each session for each rat (fullOutcomePercent).
**Figure supplement 2—source data 3.** Statistics.

kinematic changes with 'during reach' stimulation not observed with 'between reach' stimulation. This served as a positive control and reinforces the importance of the timing of dopamine neuron stimulation with respect to specific actions.

While dopamine neuron inhibition during reaching did not affect success rate (*Figure 4C*), it caused subtle changes in reach-to-grasp kinematics. Maximum reach extent lengthened slightly under dopamine neuron inhibition (that is, the paw extended further past the pellet, *Figure 7A,C*,

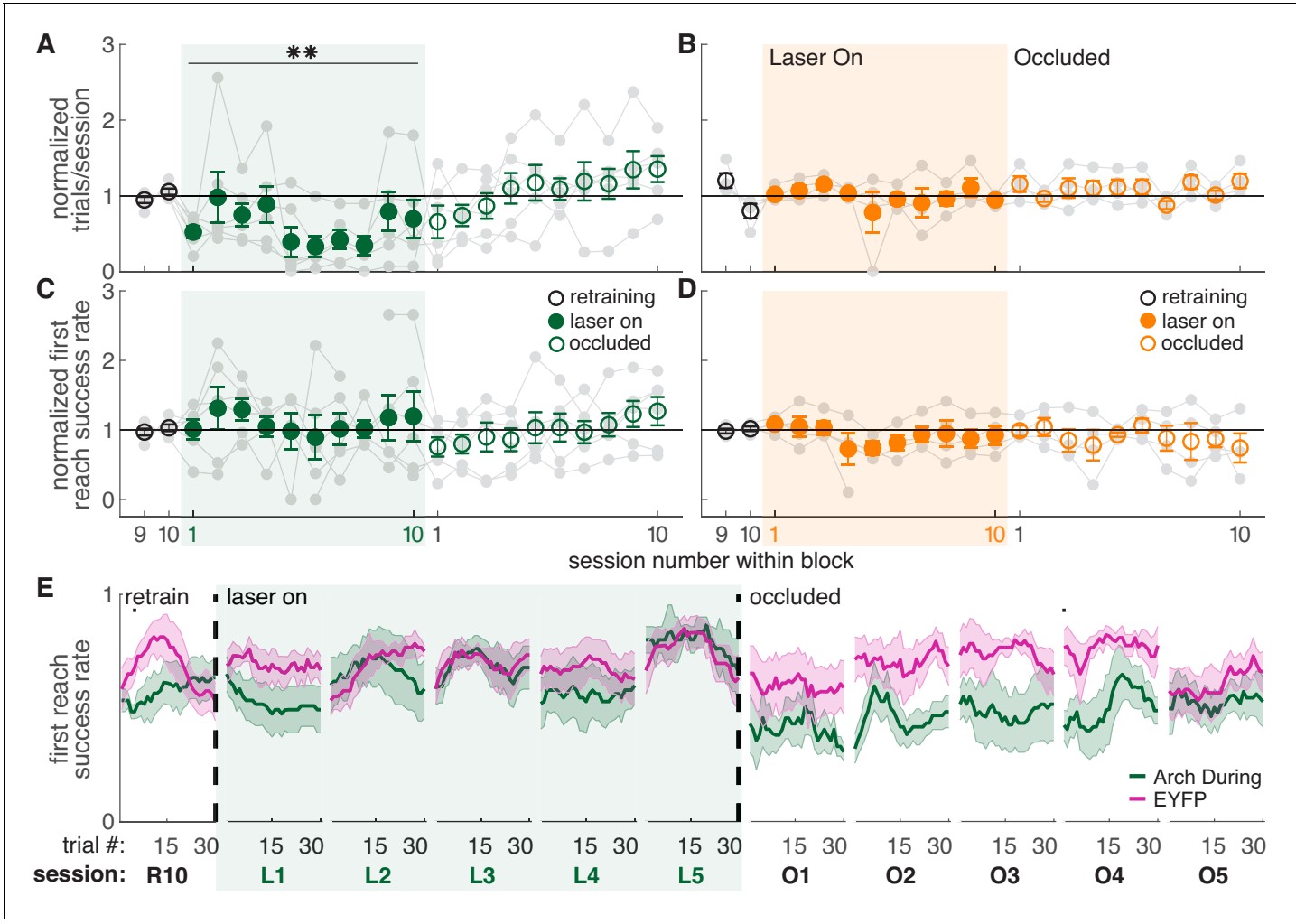

**Figure 4.** Dopamine neuron inhibition during reaches decreases the number of reaches performed but does not impair reach accuracy. (A) Average number of trials per session divided by the baseline number of trials for 'during reach' inhibition. Gray lines represent individual rats. Linear mixed model: effect of laser: $t(80) = -0.21$, p=0.84; interaction between laser and session: $t(584) = -2.64$, p=8.47×10$^{-3}$. (B) Same as (A) for 'between reach' inhibition. Linear mixed model: effect of laser: $t(80) = 0.93$, p=0.36; interaction between laser and session: $t(584) = -1.52$, p=0.13. (C) Average first attempt success rate divided by baseline success rate for 'during reach' inhibition. Linear mixed model: effect of laser: $t(133) = 0.59$, p=0.56; interaction between laser and session: $t(584) = -0.40$, p=0.69. (D) Same as (C) for 'between reach' inhibition. Linear mixed model: effect of laser: $t(133) = -0.64$, p=0.52; interaction between laser and session: $t(584) = 0.10$, p=0.92. (E) Moving average of success rate across trials within individual sessions in the last retraining session, first 5 'laser on' sessions, and first 5 'occlusion' sessions. Unlike panels C and D, these are not normalized to retraining sessions because they are within-session moving averages. Black bars represent statistically significant differences between groups (Wilcoxon rank sum test, p<0.01). 'R10', 'L1', 'O1', etc. indicate the 10th retraining session, first 'laser on' session, first 'occlusion' session, etc. *Figure 4—figure supplement 3* shows additional performance measures for 'during reach' and 'between reach' Arch activation ('any reach' success rates, number of attempts per trial, and breakdown of reach outcomes across sessions). Shaded colored areas in (E) and error bars in A-D represent s.e.m. Moving average of success rate within sessions for Arch Between rats is shown in *Figure 4—figure supplement 1*. Data for individual Arch During and Arch Between rats are shown in *Figure 4—figure supplement 2*. ** indicates p<0.01 for the laser-session interaction term in panel A.

The online version of this article includes the following source data and figure supplement(s) for figure 4:

**Source data 1.** A .mat file containing number of trials (num_trials) and first reach success rate (firstReachSuccess) for 22 testing sessions ('retraining', 'laser on', and 'occluded').

**Source data 2.** A .mat file containing first reach success rate averages across a moving block of 10 trials for 22 testing sessions ('retraining', 'laser on', and 'occluded').

**Source data 3.** Statistics.

**Figure supplement 1.** Moving average of success rate across trials within individual sessions in Arch-injected rats receiving dopamine neuron inhibition between reaches.

**Figure supplement 1—source data 1.** A .mat file containing first reach success rate averages across a moving block of 10 trials for 22 testing sessions ('retraining', 'laser on', and 'occluded').

*Figure 4 continued on next page*

*Figure 4 continued*

**Figure supplement 1—source data 2.** Statistics.

**Figure supplement 2.** Individual rat data for moving average of success rate across trials within individual sessions ('Arch During' and 'Arch Between').

**Figure supplement 2—source data 1.** A .mat file containing first reach success rate averages across a moving block of 10 trials for 22 testing sessions ('retraining', 'laser on', and 'occluded').

**Figure supplement 3.** Additional task performance measures for 'during reach' and 'between reach' Arch activation.

**Figure supplement 3—source data 1.** A .mat file containing any reach success rate (anyReachSuccess) and number of reach attempts (mean_num_reaches) for 22 testing sessions ('retraining', 'laser on', and 'occluded').

**Figure supplement 3—source data 2.** A .mat file containing percentage of trials that were each outcome type for each session for each rat (fullOutcomePercent).

**Figure supplement 3—source data 3.** Statistics.

*Videos 3* and *4*), in opposition to the effects of dopamine neuron stimulation. This effect almost reached significance in the linear mixed-effect model (p=0.091, see *Figure 7* caption), but a contrast test comparing laser day 10 to occlusion day one was significant ($t(37) = -3.24$, p=0.003). Furthermore, reach extent consistently lengthened at the individual rat level (*Figure 7A*, gray markers) as well as across trials within sessions (*Figure 7C*, *Figure 7—figure supplement 3*). Maximum reach velocity also decreased with dopamine inhibition (*Figure 7H,I*). This was not quite significant in the linear mixed-effect model (p=0.094, see *Figure 7* caption), but there was a significant difference between laser day 10 and occlusion day 1 (contrast testing, $t(33) = -2.49$, p=0.018). These data suggest that dopamine neuron stimulation and inhibition have roughly opposite effects on reach kinematics. Dopamine neuron inhibition did not significantly affect grasp aperture (*Figure 7D,E*) or paw orientation (*Figure 7F,G*), potentially due to ceiling effects. None of these measures differed between successful and failed reaches (*Figure 7—figure supplement 7*). No kinematic changes were observed in rats that received dopamine neuron inhibition between reaches (*Figure 7B* and *Figure 7—figure supplements 2–8*).

## Dopamine manipulations disrupt reach-to-grasp coordination

Reach-to-grasp success requires precise coordination of a complex sequence of reach sub-movements. Reaches begin when the rat orients to the pellet with its nose, then lifts and aligns its paw at midline with the digits closed. As the forelimb advances towards the pellet, the digits extend and spread while the paw pronates. After the digits close to grasp the pellet, the forelimb and paw are raised and supinated to bring the pellet toward the mouth (*Alaverdashvili and Whishaw, 2010*; *Whishaw and Pellis, 1990*; *Whishaw et al., 2008*). Because fine motor coordination is impaired in patients with PD, including during reaching-to-grasp (*Whishaw et al., 2002*), we looked to see if the coordination of reach sub-movements was affected by dopamine neuron stimulation or inhibition.

Dopamine neuron stimulation during reaching altered the coordination of digit spread (aperture) and paw pronation (orientation) with respect to paw advancement (*Figures 8,9*). Aperture increased earlier (when the paw was further from the pellet) in 'during reach' stimulation sessions compared to 'retraining' or 'occlusion' sessions (*Figure 9A,B*). Thus, during dopamine stimulation, aperture was smaller at reach end but larger (on average) at matched distances from the pellet (*Figures 6C,8,9E*). 'During reach' dopamine neuron inhibition had the opposite effect – paw aperture began to increase when the paw was closer to the pellet compared to 'retraining' or 'occlusion' sessions (*Figure 9A,B*). Similar changes occurred with paw orientation: during sessions with dopamine neuron stimulation, paw pronation began further from the pellet (*Figure 9D,E,F*). Dopamine neuron inhibition, however, did not affect the relationship between paw orientation and paw advancement. As for other kinematic effects, the changes in coordination progressed across sessions (most evident in *Figure 9C,F*). No changes were observed in rats that received dopamine stimulation or inhibition between reaches or in EYFP control rats (*Figure 9B,C,E,F* and *Figure 9—figure supplements 1–5*). Together, these results suggest that dopamine neuron stimulation accelerates transitions between reach sub-movements, while dopamine neuron inhibition has the opposite effect.

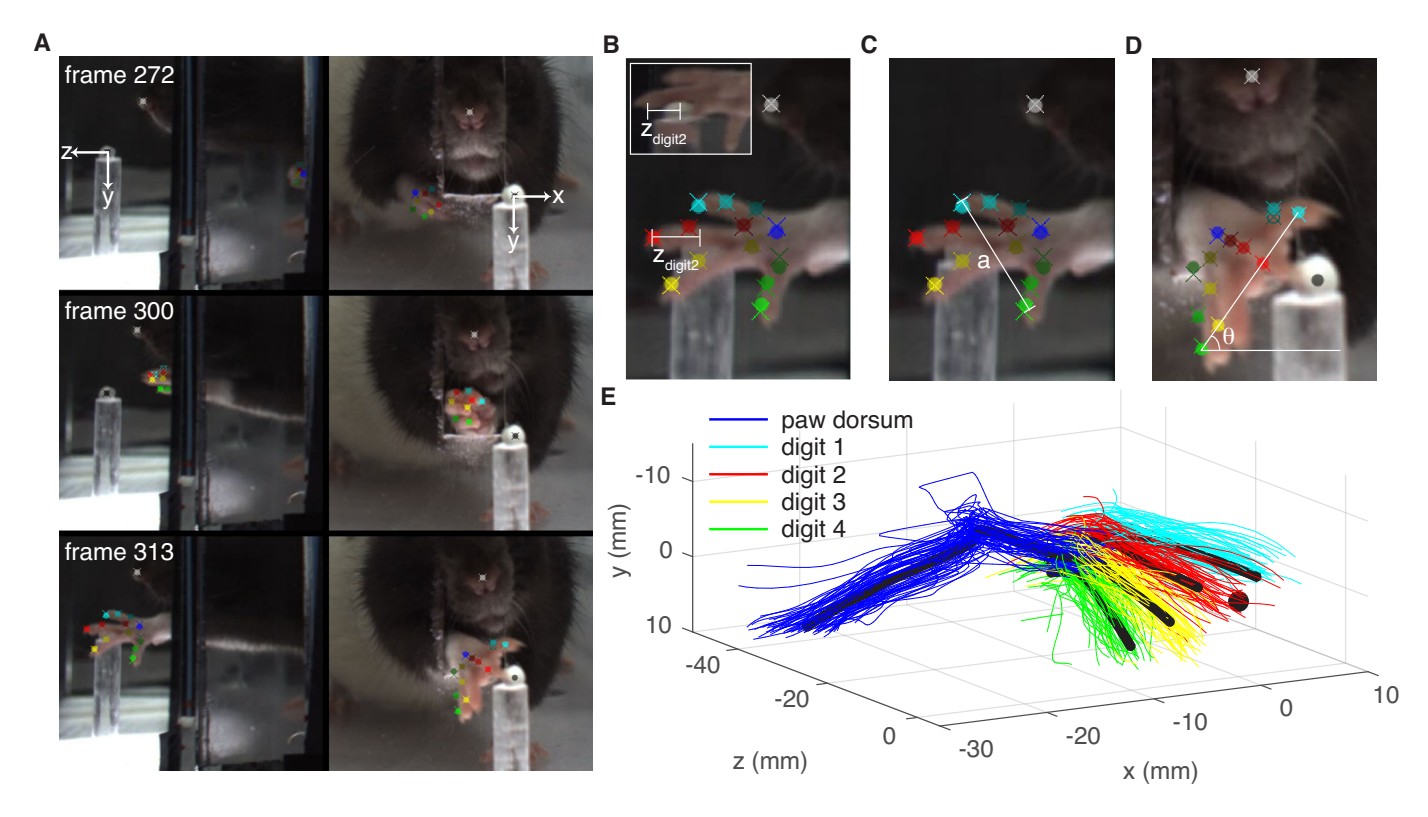

**Figure 5.** Paw and digit tracking with Deeplabcut. (A) Deeplabcut identification of digits, paw dorsum, nose, and pellet in individual video frames (side mirror and direct views). X, Y, and Z coordinates are in reference to the pellet. (B) Reach extent ($z_{digit2}$) is the z-coordinate of the tip of the second digit. The end of a reach is defined as the moment $z_{digit2}$ begins to decrease (the digit tip moves back toward the box). Inset – mirror view of the palmar surface of the paw (C) Grasp aperture (a) is the Euclidian distance between the first and fourth digit tips. (D) Paw orientation is the angle (θ) between a line connecting the first and fourth digit tips and the floor. (E) Example three-dimensional reconstruction of reaching trajectories from a single 'retraining' session. Colored lines represent individual trials and black lines represent average trajectories of the paw dorsum and digit tips. Sugar pellet (black dot) is at (0,0,0).

The online version of this article includes the following source data for figure 5:

**Source data 1.** A .mat file containing paw trajectories (reachData.pd_trajectory) and individual digit trajectories (reachData.dig_trajectory) for each trial in a single 'retraining' session for a 'ChR2 During' rat.

**Source data 2.** A .mat file containing average paw (ratSummary.mean_pd_trajectory) and digit (ratSummary.mean_dig_trajectories) trajectories across all 22 testing sessions ('retraining', 'laser on', and 'occluded') for the same rat as in *Figure 5—source data 1*.

## Dopamine neuron stimulation establishes distinct reach-to-grasp representations

Dopamine neuron stimulation gradually induced changes in reach-to-grasp kinematics, but kinematics rapidly recovered to baseline when the laser was occluded. We next asked if reinstating dopamine neuron stimulation would again gradually alter reach kinematics. Following testing with the laser occluded, ChR2-injected rats that had received 'between reach' stimulation performed additional 'during reach' stimulation sessions (*Figure 10A*). These continued until reach-to-grasp kinematics were impaired (average: 3.17 ± 0.98 sessions) (*Figure 10B, C*, *Figure 10—figure supplements 1,2*). Once kinematics were impaired, rats performed an additional one or two 30-min sessions during which the laser alternated every five trials between being off and on during reaches (*Figure 10D*).

Rats transitioned rapidly between 'normal' and 'impaired' reach-to-grasp kinematics with the laser off and on, respectively. The mean success rate dropped within a single trial of laser stimulation

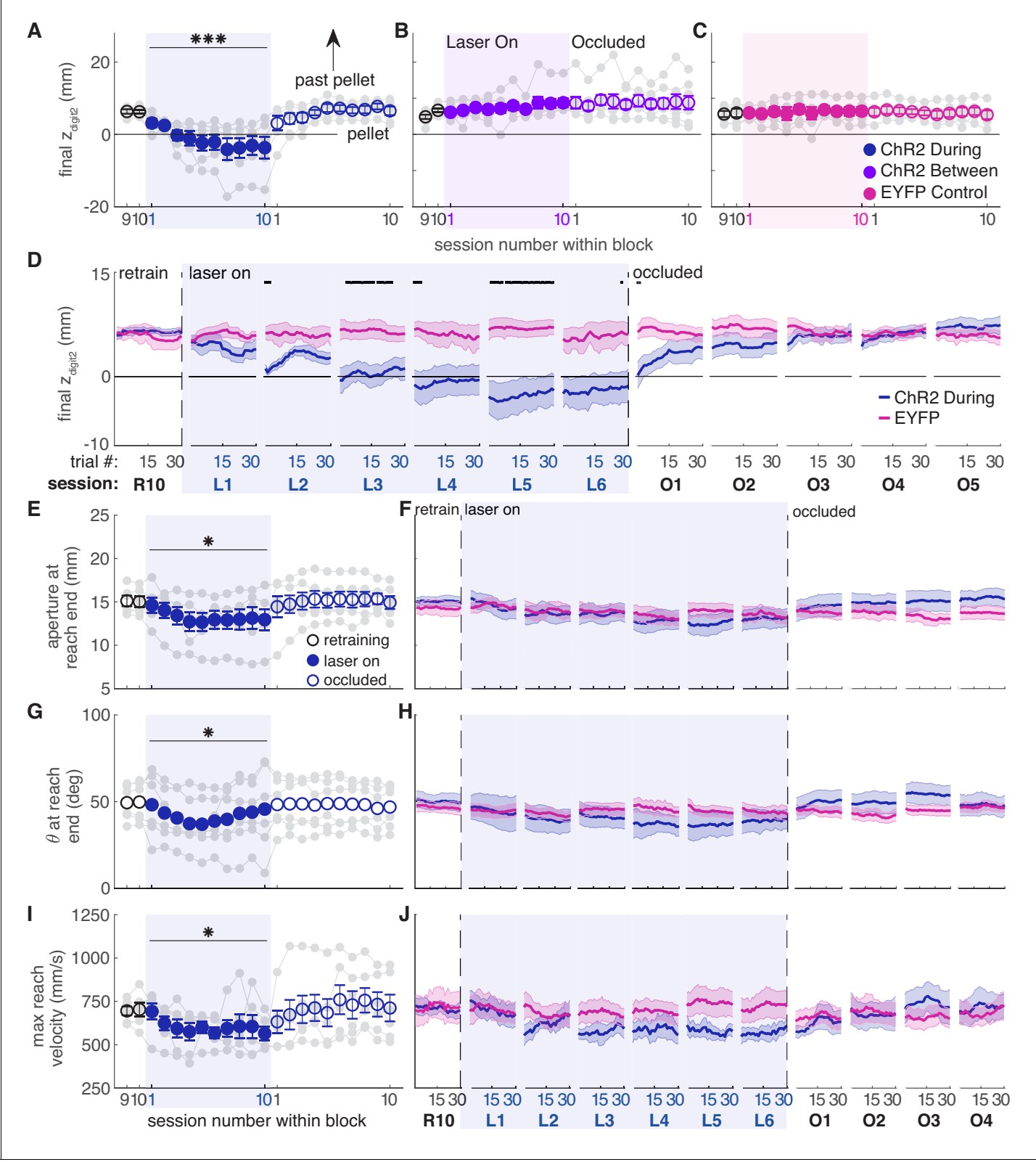

**Figure 6.** Dopamine neuron stimulation induces progressive changes in reach-to-grasp kinematics. (**A**) The average maximum reach extent progressively decreased across sessions with 'during reach' stimulation. Linear mixed model: effect of laser: $t(62) = 1.70$, p=0.09; interaction between laser and session: $t(585) = 6.88$, p=$1.59\times10^{-11}$. Average maximum reach extent returned to baseline within the first 'occlusion' session. Contrast testing ('retraining' session 10 vs. 'occlusion' session 1): $t(585) = 1.62$, p=0.11. (**B**) Same as (**A**) for 'between reach' stimulation. Linear mixed model: effect of

*Figure 6 continued on next page*

*Figure 6 continued*

laser: $t(62) = 0.02$, p=0.99; interaction between laser and session: $t(585) = -0.43$, p=0.67. (C) Same as (A) and (B) for 'during reach' illumination in control EYFP-injected rats. Linear mixed model: effect of laser: $t(62) = 0.10$, p=0.92; interaction between laser and session: $t(585) = -0.68$, p=0.50. *Figure 6—figure supplement 1* shows the relationship between reach extent and fiber tip location for these groups. (D) Moving average of maximum reach extent within the last 'retraining' session, first 6 'laser on' sessions, and first 5 'occlusion' sessions. Black bars represent trials with a statistically significant difference between groups (Wilcoxon rank sum test, p<0.01). (E) Average grasp aperture at reach end for 'during reach' stimulation. Linear mixed model: effect of laser: $t(48) = -1.34$, p=0.19; interaction between laser and session: $t(585) = -2.19$, p=0.03. Average aperture returned to baseline within the first 'occlusion' session. Contrast testing ('retraining' session 10 vs. 'occlusion' session 1): $t(585) = -0.87$, p=0.38. (F) Moving average of aperture at reach end within the last 'retraining' session, first 6 'laser on' sessions, and first 4 'occlusion' sessions. (G) Same as (E) for paw orientation. Linear mixed model: effect of laser: $t(74) = -2.52$, p=0.01; interaction between laser and session: $t(585) = 0.19$, p=0.85. Average angle returned to baseline within the first 'occlusion' session. Contrast testing ('retraining' session 10 vs. 'occlusion' session 1): $t(585) = 1.64$, p=0.10. (H) Moving average of paw angle at reach end across trials in the last (10th) 'retraining' session, first 6 'laser on' sessions, and first 4 'occlusion' sessions. (I) Same as (E) and (G) for maximum reach velocity. Linear mixed model: effect of laser: $t(49) = -0.45$, p=0.65; interaction between laser and session: $t(585) = -2.45$, p=0.01. Average velocity returned to baseline within the first 'occlusion' session. Contrast testing ('retraining' session 10 vs. 'occlusion' session 1): $t(585) = -1.64$, p=0.10. (J) Moving average of maximum reach velocity within the last 'retraining' session, first 6 'laser on' sessions, and first 4 'occlusion' sessions. 'R10', 'L1', 'O1', etc. indicate the 10th retraining session, first 'laser on' session, first 'occlusion' session, etc. Shaded colored areas in D, F, H, J and error bars in A, B, C, E, I represent s.e.m. Similar data for ChR2 Between rats are shown in *Figure 6—figure supplement 2*. Individual rat data are shown in *Figure 6—figure supplements 3–6*. *Figure 6—figure supplement 7* compares kinematic measures (final $z_{digit2}$, aperture, paw orientation, and maximum reach velocity) between successful and unsuccessful reaches for ChR2 during, ChR2 between, and EYFP rats. *Figure 6—figure supplement 8* shows success rate and kinematic measures as a function of final $z_{digit2}$ for ChR2 during, ChR2 between, and EYFP rats. * indicates p<0.05 for the laser or laser-session interaction terms in panels E, G, I. *** indicates $p < 1.0 \times 10^{-10}$ for the laser-session interaction term in panel A. The online version of this article includes the following source data and figure supplement(s) for figure 6:

**Source data 1.** A .mat file containing maximum reach extent of digit2 (mean_dig2_endPt), aperture at reach end (mean_end_aperture), paw orientation at reach end (mean_end_orientations), and mean paw velocity (mean_pd_v) for 22 testing sessions ('retraining', 'laser on', and 'occluded').
**Source data 2.** A .mat file containing digit2 endpoint (digEnd), aperture at reach end (aperture), orientation at reach end (orientation), and velocity (velocity) averages across a moving block of 10 trials for 22 testing sessions ('retraining', 'laser on', and 'occluded').
**Source data 3.** Statistics.
**Figure supplement 1.** Maximum reach extent on laser day 10 as a function of anterior-posterior, medial-lateral, and dorsal-ventral fiber tip coordinates referenced to bregma (A–P and M–L) or brain surface (D–V).
**Figure supplement 1—source data 1.** A .mat file containing digit2 endpoints (mean_dig2_endPt) for 22 testing sessions ('retraining', 'laser on', and 'occluded').
**Figure supplement 1—source data 2.** A .xlsx file containing the estimated fiber tip locations for each rat in all groups.
**Figure supplement 2.** Reach-to-grasp kinematics do not change in EYFP control rats or ChR2-injected rats receiving between reach stimulation.
**Figure supplement 2—source data 1.** A .mat file containing aperture at reach end (mean_end_aperture), paw orientation at reach end (mean_end_orientations), and mean paw velocity (mean_pd_v) for 22 testing sessions ('retraining', 'laser on', and 'occluded').
**Figure supplement 2—source data 2.** A .xlsx file containing the estimated fiber tip locations for each rat in all groups.
**Figure supplement 2—source data 3.** Statistics.
**Figure supplement 3.** Individual rat data for moving average of maximum reach extent across trials within individual sessions ('ChR2 During', 'ChR2 Between' and 'EYFP').
**Figure supplement 3—source data 1.** A .mat file containing digit2 endpoint (digEnd) averages across a moving block of 10 trials for 22 testing sessions ('retraining', 'laser on', and 'occluded').
**Figure supplement 4.** Individual rat data for moving average of grasp aperture across trials within individual sessions ('ChR2 During', 'ChR2 Between' and 'EYFP').
**Figure supplement 4—source data 1.** A .mat file containing aperture at reach end averages across a moving block of 10 trials for 22 testing sessions ('retraining', 'laser on', and 'occluded').
**Figure supplement 5.** Individual rat data for moving average of paw angle across trials within individual sessions ('ChR2 During', 'ChR2 Between' and 'EYFP').
**Figure supplement 5—source data 1.** A .mat file containing orientation at reach end averages across a moving block of 10 trials for 22 testing sessions ('retraining', 'laser on', and 'occluded').
**Figure supplement 6.** Individual rat data for moving average of maximum reach velocity across trials within individual sessions ('ChR2 During', 'ChR2 Between' and 'EYFP').
**Figure supplement 6—source data 1.** A .mat file containing reach velocity averages across a moving block of 10 trials for 22 testing sessions ('retraining', 'laser on', and 'occluded').
**Figure supplement 7.** Kinematic measures separated by reach success or failure for ChR2 During (left column), ChR2 Between (middle column), and EYFP (right column) groups.
**Figure supplement 7—source data 1.** A .mat file containing digit2 endpoint (mean_dig2_endPt_z), aperture at reach end (mean_end_aperture), orientation at reach end (mean_end_orientation), and reach velocity (mean_pd_v) separated by trial outcome.

*Figure 6 continued on next page*

*Figure 6 continued*

**Figure supplement 8.** Differences between reaches matched for reach extent in the ChR2 during (left column), ChR2 between (middle column), and EYFP (right column) groups.

**Figure supplement 8—source data 1.** A .mat file containing digit2 endpoint (digEnd), aperture at reach end (aperture), orientation at reach end (orient), and trial outcomes (outcome) for all trials across all sessions for each rat.

**Figure supplement 8—source data 2.** Statistics.

and improved within one trial when laser stimulation was removed (***Figure 10F***, ***Video 5***). Similarly, reach kinematics required only one trial to switch between normal and aberrant reaching patterns. Laser stimulation at the beginning of an 'on' block (trial 1) caused immediate decreases in maximum reach extent and digit aperture, which remained steady for the remaining 'Laser On' trials. Similarly, maximum reach extent and digit aperture immediately increased upon cessation of dopamine neuron stimulation (trial 1, 'Laser Off') and remained steady throughout the 'Laser Off' block (***Figure 10G,H***). There was also a significant change in paw orientation at reach end with dopamine neuron stimulation (***Figure 10I***). However, pronation decreased in these rats unlike in the 'ChR2 During' group (***Figure 6G***). There was no significant difference in maximum reach velocity between 'Laser On' and 'Laser Off' blocks (***Figure 10J***). These data indicate that once distinct reaching kinematics have been established by repeated dopaminergic manipulations, current reach kinematics are determined by the activity of nigral dopamine neurons on that trial.

## Dopamine neuron stimulation induces context- and history-dependent abnormal involuntary movements

To verify fiber placement and opsin expression prior to reaching experiments, we placed rats in a clear cylinder and illuminated SNc with blue light of varying intensity (***Figure 11***). We predicted that rats with well-placed fibers expressing high levels of ChR2 would develop increasingly worse abnormal involuntary movements (AIMs) as laser intensity increased. To our surprise, rats that subsequently developed markedly abnormal reach kinematics during the skilled reaching task appeared unaffected by dopamine neuron stimulation in the cylinder (AIMs Test 1, ***Figure 11***). In 'post-reaching' cylinder sessions (AIMS Test 2, ***Figure 11***), however, dopamine neuron stimulation elicited markedly abnormal movements (***Figure 11***, ***Video 6***). While AIMs were obvious in the context of the cylinder, the same (or higher) stimulation intensities delivered while rats were reaching rarely elicited abnormal movements (other than altered reach kinematics, ***Figure 11A***, ***Video 7***). Only three of six 'ChR2 during' rats exhibited any AIMs superimposed on reaches during 'laser on' day 10 (2 rats each with AIMs in 1 of 10 reaches, one rat with AIMs in 5 of 10 reaches), and none exhibited dyskinesias during reaching on laser day 2 when success rate was already near zero. Thus, while dyskinesias likely accounted for some reaching deficits, that was not the primary failure mechanism. Furthermore, rats that did not experience dyskinesias in the cylinder still had markedly reduced reaching success, and

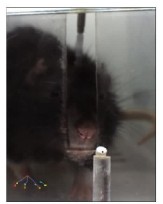

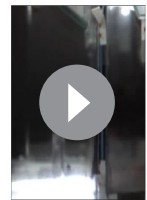

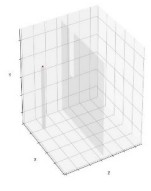

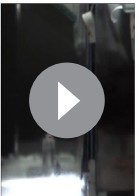

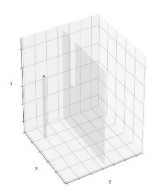

**Video 1.** Sample reach during the last 'retraining' session of a 'ChR2-During' rat showing the direct camera view, the mirror view of the paw dorsum, and 3D skeleton reconstruction. Two trailing points are shown for each body part/object. Video is slowed 10x.
https://elifesciences.org/articles/61591#video1

**Video 2.** Sample reach during the seventh 'laser on' session for the same rat as in ***Video 1*** showing the direct camera view, the mirror view of the paw dorsum, and 3D skeleton reconstruction. Two trailing points are shown for each body part/object. Video is slowed 10x.
https://elifesciences.org/articles/61591#video2

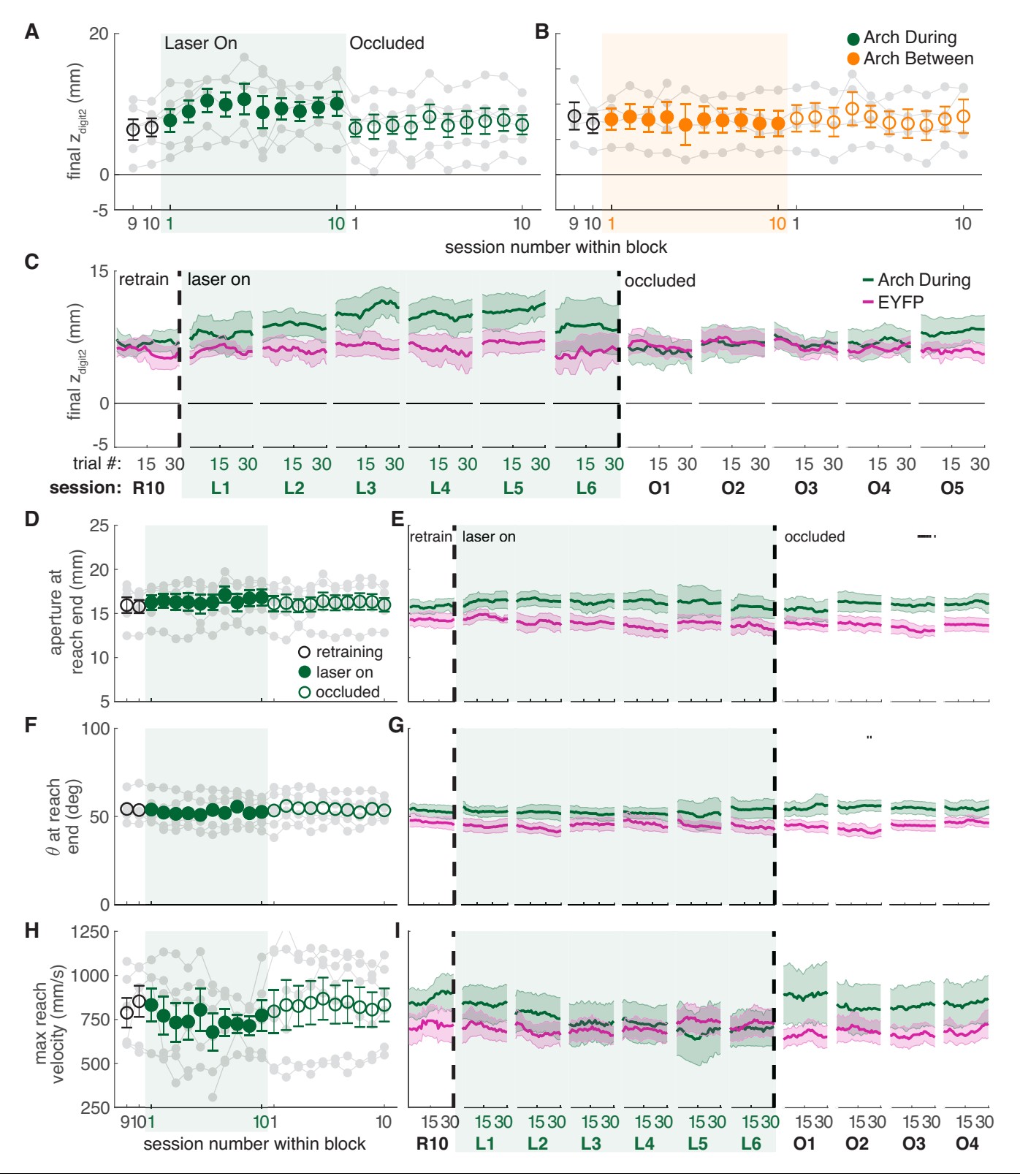

**Figure 7.** Dopamine neuron inhibition induces subtle changes in reach-to-grasp kinematics. (A) Average maximum reach extent across sessions for 'during reach' inhibition. Linear mixed model: effect of laser: $t(63) = -1.72$, p=0.09; interaction between laser and session: $t(585) = 0.03$, p=0.98. (B) Same as (A) for 'between reach' inhibition. Linear mixed model: effect of laser: $t(63) = -0.23$, p=0.82; interaction between laser and session: $t(585) = 0.99$, p=0.32. *Figure 7—figure supplement 1* shows the relationship between reach extent and fiber tip location for these groups. (C) Moving average

*Figure 7 continued on next page*

*Figure 7 continued*

of maximum reach extent within the last 'retraining' sessions, first 6 'laser on' sessions, and first 5 'occlusion' sessions. (**D**) Same as (**A**) for aperture: effect of laser: $t(48) = 0.53$, p=0.60; interaction between laser and session: $t(585) = 1.76$, p=0.08. (**E**) Moving average of grasp aperture at reach end within the last 'retraining' session, first 6 'laser on' sessions, and first 4 'occlusion' sessions. (**F**) Same as (**A**) and (**D**) for paw orientation: effect of laser: $t(75) = −0.20$, p=0.84; interaction between laser and session: $t(585) = −0.28$, p=0.78. (**G**) Moving average of paw angle at reach end within the last 'retraining' session, first 6 'laser on' sessions, and first 4 'occlusion' sessions. (**H**) Same as (**A**), (**D**), and (**F**) for maximum reach velocity: effect of laser: $t(49) = −0.52$, p=0.60; interaction between laser and session: $t(585) = −1.68$, p=0.09. (**I**) Moving average of maximum reach velocity within the last 'retraining' session, first 6 'laser on' sessions, and first 4 'occlusion' sessions. 'R10', 'L1', 'O1', etc. indicate the 10th retraining session, first 'laser on' session, first 'occlusion' session, etc. Shaded colored areas in C, E, G, I and error bars in A, B, D, H represent s.e.m. Similar data for Arch Between rats are shown in *Figure 7—figure supplement 2*. Individual rat data are shown in *Figure 7—figure supplements 3–6*. *Figure 7—figure supplement 7* compares kinematic measures (final $z_{digit2}$, aperture, paw orientation, and maximum reach velocity) between successful and unsuccessful reaches for Arch during and Arch between rats. *Figure 7—figure supplement 8* shows success rate and kinematic measures as a function of final $z_{digit2}$ for Arch during and Arch between rats. Black bars in (**E**) and (**G**) represent trials with a statistically significant difference between groups (Wilcoxon rank sum test, p<0.01).

The online version of this article includes the following source data and figure supplement(s) for figure 7:

**Source data 1.** A .mat file containing maximum reach extent of digit2 (mean_dig2_endPt), aperture at reach end (mean_end_aperture), paw orientation at reach end (mean_end_orientations), and mean paw velocity (mean_pd_v) for 22 testing sessions ('retraining', 'laser on', and 'occluded').

**Source data 2.** A .mat file containing digit2 endpoint (digEnd), aperture at reach end (aperture), orientation at reach end (orientation), and velocity (velocity) averages across a moving block of 10 trials for 22 testing sessions ('retraining', 'laser on', and 'occluded').

**Source data 3.** Statistics.

**Figure supplement 1.** Maximum reach extent on laser day 10 as a function of anterior-posterior, medial-lateral, and dorsal-ventral fiber tip coordinates referenced to bregma (**A–P and M–L**) or brain surface (**D–V**).

**Figure supplement 1—source data 1.** A .mat file containing digit2 endpoints (mean_dig2_endPt) for 22 testing sessions ('retraining', 'laser on', and 'occluded').

**Figure supplement 1—source data 2.** A .xlsx file containing the estimated fiber tip locations for each rat in all groups.

**Figure supplement 2.** Reach-to-grasp kinematics do not change with dopamine neuron inhibition between reaches.

**Figure supplement 2—source data 1.** A .mat file containing aperture at reach end (mean_end_aperture), paw orientation at reach end (mean_end_orientations), and mean paw velocity (mean_pd_v) for 22 testing sessions ('retraining', 'laser on', and 'occluded').

**Figure supplement 2—source data 2.** A .mat file containing digit2 endpoint (digEnd), aperture at reach end (aperture), orientation at reach end (orientation), and velocity (velocity) averages across a moving block of 10 trials for 22 testing sessions ('retraining', 'laser on', and 'occluded').

**Figure supplement 2—source data 3.** Statistics.

**Figure supplement 3.** Individual rat data for moving average of maximum reach extent within individual sessions ('Arch During' and 'Arch Between').

**Figure supplement 3—source data 1.** A .mat file containing digit2 endpoint (digEnd) averages across a moving block of 10 trials for 22 testing sessions ('retraining', 'laser on', and 'occluded').

**Figure supplement 4.** Individual rat data for moving average of grasp aperture at reach end within individual sessions ('Arch During' and 'Arch Between').

**Figure supplement 4—source data 1.** A .mat file containing aperture at reach end averages across a moving block of 10 trials for 22 testing sessions ('retraining', 'laser on', and 'occluded').

**Figure supplement 5.** Individual rat data for moving average of paw angle at reach end within individual sessions ('Arch During' and 'Arch Between').

**Figure supplement 5—source data 1.** A .mat file containing orientation at reach end averages across a moving block of 10 trials for 22 testing sessions ('retraining', 'laser on', and 'occluded').

**Figure supplement 6.** Individual rat data for moving average of maximum reach velocity within individual sessions ('Arch During' and 'Arch Between').

**Figure supplement 6—source data 1.** A .mat file containing reach velocity averages across a moving block of 10 trials for 22 testing sessions ('retraining', 'laser on', and 'occluded').

**Figure supplement 7.** Kinematic measures separated by reach success or failure for Arch during (left column) and Arch between (right column) groups.

**Figure supplement 7—source data 1.** A .mat file containing digit2 endpoint (mean_dig2_endPt_z), aperture at reach end (mean_end_aperture), orientation at reach end (mean_end_orientation), and reach velocity (mean_pd_v) separated by trial outcome.

**Figure supplement 8.** Differences between reaches matched for reach extent in the Arch during (left column) and Arch between (right column) groups.

**Figure supplement 8—source data 1.** A .mat file containing digit2 endpoint (digEnd), aperture at reach end (aperture), orientation at reach end (orient), and trial outcomes (outcome) for all trials across all sessions for each rat.

**Figure supplement 8—source data 2.** Statistics.

rats that experienced dyskinesias could have preserved success rates (*Figure 11—figure supplement 1*). Therefore, the expression of dopamine-dependent AIMs depends not only on current levels of dopamine neuron activation, but the history of prior activation and the current behavioral context.

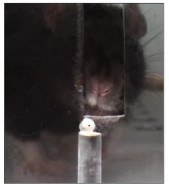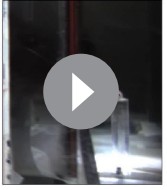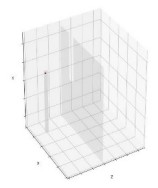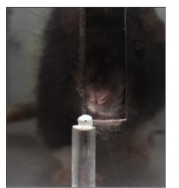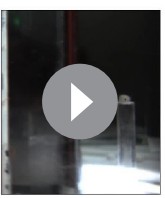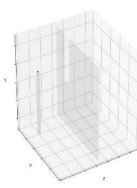

**Video 3.** Sample reach during the last 'retraining' session of an 'Arch-During' rat showing the direct camera view, the mirror view of the paw dorsum, and 3D skeleton reconstruction. Two trailing points are shown for each body part/object. Video is slowed 10x.
https://elifesciences.org/articles/61591#video3

**Video 4.** Sample reach during the tenth 'laser on' session for the same rat as in *Video 3* showing the direct camera view, the mirror view of the paw dorsum, and 3D skeleton reconstruction. While the reaches in *Video 3* and 4 are superficially similar, the rat reaches further past the pellet after repeated dopamine neuron inhibition. Two trailing points are shown for each body part/object. Video is slowed 10x.
https://elifesciences.org/articles/61591#video4

## Discussion

Our goal was to determine how midbrain dopamine neuron manipulations affect dexterous skill. Our results revealed a role for dopamine in motor learning, as repeated dopamine manipulations induced gradual changes in reach-to-grasp kinematics. These manipulations not only affected gross performance measures (e.g. velocity and amplitude) but also disrupted coordinated execution of reach sub-movements. Once dopamine stimulation-induced changes were established, reach-to-grasp kinematics depended strongly on the dopamine status of the current trial. Furthermore, these effects were temporally specific – only manipulations during reaches influenced forelimb kinematics. Finally, the effect of dopamine on motor control is context-dependent, as the same dopamine stimulation in the cylinder and reaching chamber induced distinct behavioral responses.

The history-dependent effects of dopamine on skilled reaching are superficially consistent with reinforcement learning models (*Schultz, 2019*). While most evidence for phasic dopamine signals encoding RPEs comes from paradigms in which animals choose between discrete actions (e.g. press a right or left lever), recent studies suggest that dopamine encodes RPE-like 'performance prediction errors' for complex behaviors with greater degrees of freedom (*Beeler et al., 2010*; *Gadagkar et al., 2016*). It is plausible that dopamine neuron excitation/inhibition creates an artificially reinforcing/discouraging signal that influences subsequent reaches. At baseline and in EYFP controls, failed reaches were slightly (but not significantly) shorter than successful reaches (*Figure 6—figure supplement 7*). Dopamine neuron stimulation could have gradually reinforced this subtle difference until it became measurable. On the other hand, we stimulated or inhibited dopamine neurons on every trial, so that 'short reach' and 'long reach' reinforcement should cancel each other. Furthermore, stimulation and inhibition had consistent, almost opposite effects. Since there are many possible failure mechanisms, we would expect variable and unpredictable kinematic changes if dopamine neuron stimulation was artificially reinforcing failed reaches. Together, these findings suggest that kinematic changes do not result purely from performance prediction error signals, but that dopamine intrinsically biases movement kinematics in a consistent direction. Closed loop experiments in which manipulations occur only for 'long' or 'short' reaches may definitively separate these possibilities (*Yttri and Dudman, 2016*).

Motion tracking data provide insight into the nature of an intrinsic dopamine bias. A common interpretation of dopamine's role in movement is that it regulates 'vigor,' which has been defined as the speed, frequency, and amplitude of movements (*Dudman and Krakauer, 2016*). Our dopamine manipulations influenced 'vigor' in unexpected ways: dopamine neuron stimulation decreased, and inhibition increased, movement amplitude (reach extent). Furthermore, both stimulation and inhibition decreased movement speed. These effects apparently contradict previous work directly correlating dopaminergic tone with movement velocity and/or amplitude (*Carr and White, 1987*; *Leventhal et al., 2014*; *Panigrahi et al., 2015*).

This discrepancy may be due to different demands on the motor system. 'Vigor' assays generally demand movement along one dimension. For example, mice manipulating a joystick (*Panigrahi et al., 2015*), or humans moving a manipulandum to a target (*Baraduc et al., 2013*;

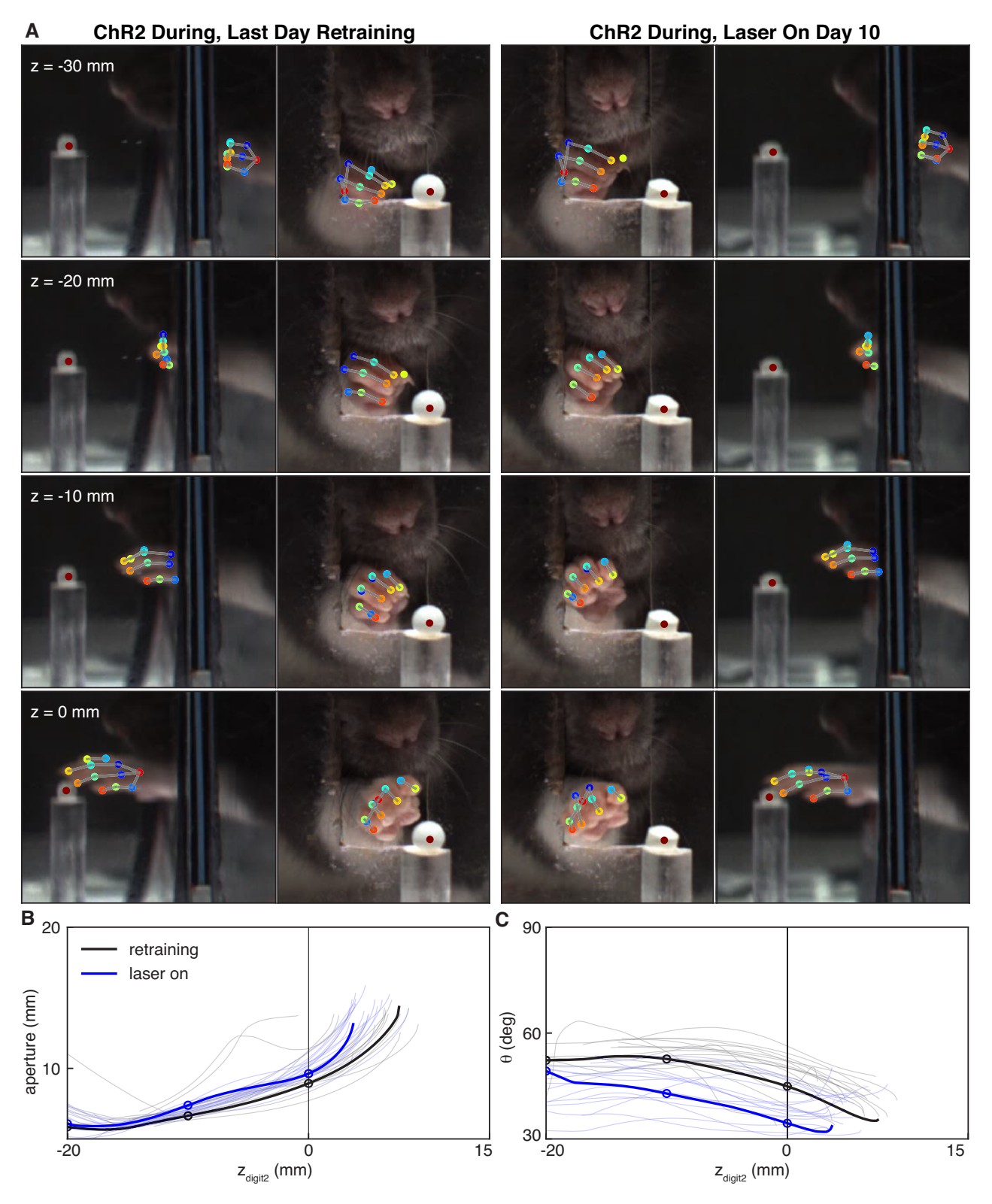

**Figure 8.** Dopamine neuron stimulation alters the coordination between digit movements and paw advancement. (A) Sample frames from single reaches at the end of 'retraining' and 'laser on' sessions from the same rat. Outer columns show the mirror views corresponding to the direct camera views in the inside columns. After 10 days of 'during reach' stimulation, the rat pronates its paw and spreads its digits further from the pellet as the paw advances. (B) Aperture as a function of the z-coordinate of the second digit tip. Solid black and blue lines correspond to the reaches shown in (A). Thin

*Figure 8 continued on next page*

*Figure 8 continued*

black and blue lines are the traces for other reaches in the same sessions. Circles indicate apertures at the corresponding $z_{digit2}$ values in (**A**). (**C**) Same as (**B**) but for paw orientation.

The online version of this article includes the following source data for figure 8:

**Source data 1.** A zip file containing .mat files to produce *Figure 8*.

*Mazzoni et al., 2007*) make forelimb/arm movements across large joints more or less along a single vector. In such tasks, dopamine-depleted subjects consistently make hypometric, bradykinetic movements. In contrast, skilled reaching comprises a sequence of precisely coordinated submovements (*Klein and Dunnett, 2012*). Stimulation caused paw pronation and digit spread to occur earlier along the reach trajectory (farther from the pellet), while inhibition delayed digit spread. Similarly, humans with PD reshape their hands closer to the target than control subjects (*Rand et al., 2006*; *Whishaw et al., 2002*). This is consistent with dopamine regulating initiation of, and transitions between, movements (*da Silva et al., 2018*; *Hamilos et al., 2020*; *Howe and Dombeck, 2016*). Within a 'vigor' framework, dopamine may invigorate the next submovement at the expense of the current one, compressing the overall sequence. In the limiting case, overlapping submovements could manifest as muscle cocontractions and dystonia, a possibility supported by the rare occurrence of abnormal movements intruding into reaches in the most severely affected rats. Simultaneous EMG recordings from multiple muscles could address this possibility (*Hyland and Jordan, 1997*).

The mechanisms by which forelimb-digit coordination occurs normally are unclear, let alone under dopamine perturbations (*Leib et al., 2020*). For ballistic movements, it is suggested that internal models predict consequences of motor commands to allow rapid feedback for online corrections (*Azim and Alstermark, 2015*). Such models necessarily integrate sensory feedback (*Azim et al., 2014*) with the passage of time (*Crevecoeur and Gevers, 2019*). Dopamine and striatal circuitry have been implicated in time estimation, though on a slower timescale than skilled reaching (*Hamilos et al., 2020*; *Mello et al., 2015*). Accelerated time perception (i.e. perceiving more time has passed than actually has) should cause premature submovement transitions, although some studies indicate that dopamine alters time perception in the opposite direction (*Soares et al., 2016*). Others suggest that dopamine regulates integration of multiple information streams (including time) for decision making (*Howard et al., 2017*). It is therefore plausible that dopamine alters reliance on sensory feedback, time perception, the internal model itself, or a combination of these elements.

Kinematic changes developed gradually, but once established they depended on the dopamine status of the current trial (*Figure 10*). Rats previously stimulated during reaching experienced larger trial-by-trial stimulation-dependent changes in reach kinematics than stimulation-naive rats or rats that had only received 'between-reach' stimulation (*Figure 10*, *Video 5*). This 'skilled reaching sensitization' shares attributes with sensitization to dopaminergic drugs (e.g. amphetamine, cocaine, and levodopa), in which repeated administration renders rodents hypersensitive to subsequent doses (*Robinson and Becker, 1986*). While the mechanisms of sensitization to dopaminergic drugs are not fully understood, enhanced dopaminergic and glutamatergic signaling are implicated (*Cadoni et al., 2000*; *Zhang et al., 2001*). Similar mechanisms could mediate skilled reaching sensitization, possibly in dorsal 'motor' striatum.

An important difference between sensitization to drugs and optogenetic stimulation is the specificity of the latter to movements performed during stimulation. Rats that had been stimulated between reaches were initially unaffected by 'during reach' stimulation (*Figure 10A–C*, *Figure 10—figure supplements 1*,*2*, *Video 5*), arguing against a general sensitizing effect. Cortico- or thalamo-striatal transmission may be sensitized specifically at synapses active during elevated dopamine signaling (*Yttri and Dudman, 2018*). Consistent with this idea, subpopulations of direct pathway medium spiny neurons are associated with dyskinesias after levodopa treatment in dopamine-depleted mice (*Ryan et al., 2018*), and selective activation of these direct pathway MSNs induces dyskinesias (*Girasole et al., 2018*). These results suggest that subpopulations of striatal output neurons encode specific dopamine-sensitive movements. In our experiments, the specific population of 'sensitized' MSNs would be those active during reaching.

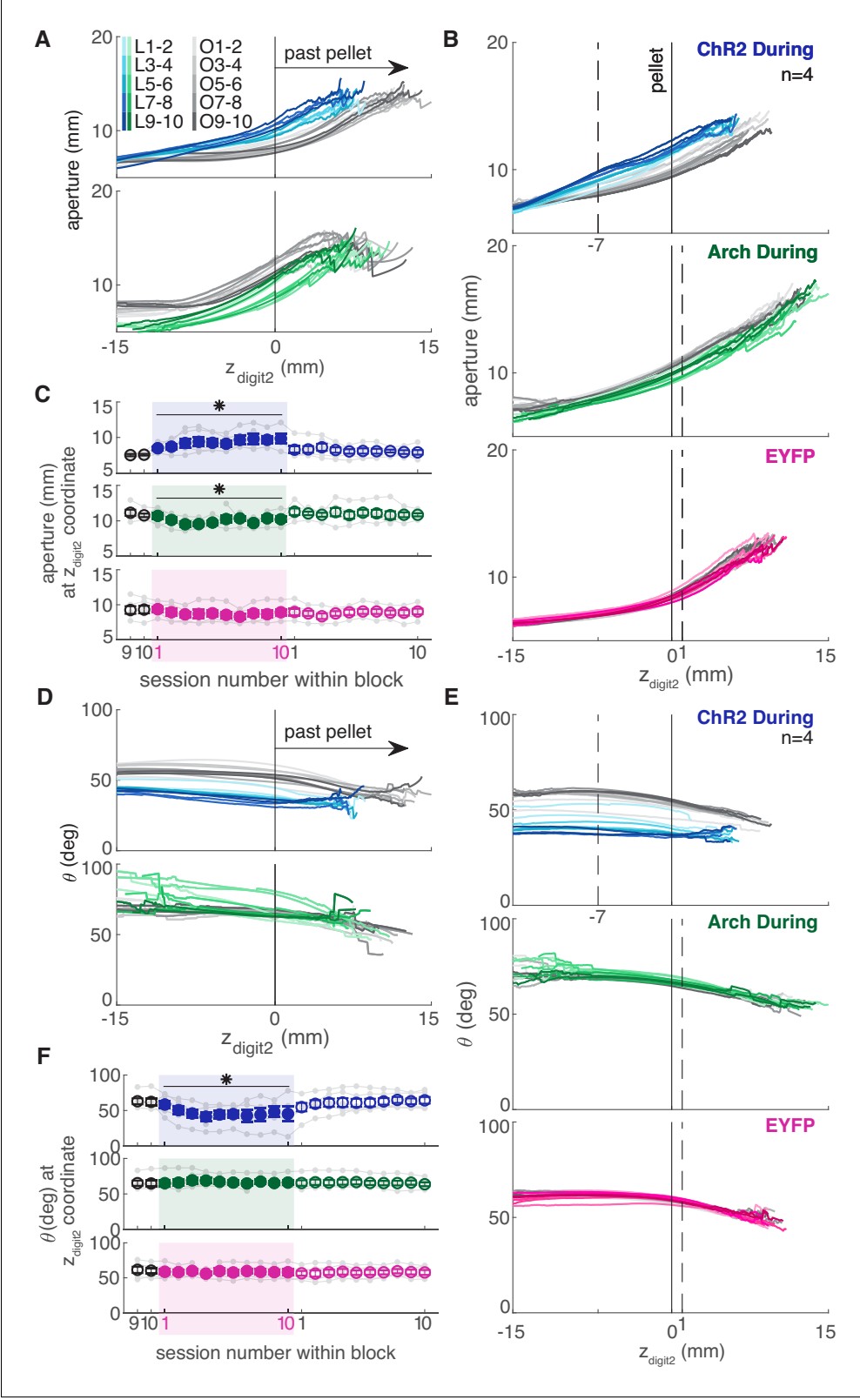

**Figure 9.** Dopamine neuron manipulations disrupt coordination of reach-to-grasp movements. (**A**) Mean aperture as a function of paw advancement ($z_{digit2}$, pellet at $z_{digit2}$ = 0) across 'laser on' and 'occlusion' sessions for exemplar rats. L1-2, O1-2, . . . indicate laser on sessions 1–2, occlusion sessions 1–2, etc. (**B**) Mean aperture as a function of paw advancement across 'laser on' and 'occlusion' sessions averaged across rats. 4 of 6 'ChR2 During' rats are included because two rats' reaches were too short in several sessions to produce a meaningful average (the average for all 6 ChR2 During rats,

*Figure 9 continued on next page*

*Figure 9 continued*

ChR2 Between rats, and Arch Between rats are shown in *Figure 9—figure supplement 1*). All rats were included for other groups. Dashed lines indicate the $z_{digit2}$ coordinate where data are sampled in (C). A more proximal $z_{digit2}$ was chosen for 'ChR2 During' because the majority of 'laser on' reaches for this group did not extend past $z_{digit2}$ = +1 mm. (C) Average grasp aperture at the $z_{digit2}$ coordinates indicated by the dashed lines in (B) across sessions. 'During reach' stimulation gradually increased aperture at 7 mm from the pellet (linear mixed model including all 6 'during reach' rats: effect of laser: $t(607)$ = 2.39, p=0.02; interaction between laser and session: $t(607)$ = 2.40, p=0.02. 'During reach' inhibition decreased aperture at 1 mm past the pellet (linear mixed model: effect of laser: $t(607)$ = −2.04, p=0.04; interaction between laser and session: $t(607)$ = 0.67, p=0.51). SNc illumination in EYFP-injected rats had no effect on aperture at 1 mm past the pellet (linear mixed model: effect of laser: $t(607)$ = −0.57, p=0.57; interaction between laser and session: $t(607)$ = −0.61, p=0.54). Gray points indicate data from individual rats. (D) Mean paw orientation as a function of paw advancement towards the pellet across 'laser on' and 'occlusion' sessions for exemplar rats. All rats are shown in *Figure 9—figure supplement 4*. (E) Mean paw orientation as a function of paw advancement across 'laser on' and 'occlusion' sessions averaged across rats. Dashed lines indicate $z_{digit2}$ coordinates where data are sampled in (F) for each group. Four of 6 'ChR2 During' rats are included because two rats' reaches were too short in several sessions to produce a meaningful average (the average for all 6 ChR2 During rats, ChR2 Between rats, and Arch Between rats are shown in *Figure 9—figure supplement 1*). (F) Average paw orientation at $z_{digit2}$ coordinates indicated by dashed lines in (E) across all sessions. 'During reach' stimulation caused a gradual increase in pronation (i.e. a smaller angle) at 7 mm from the pellet (linear mixed model including all 6 'during reach' rats: effect of laser: $t(607)$ = −2.34, p=0.02; interaction between laser and session: $t(607)$ = −2.33, p=0.02). 'During reach' inhibition had no effect on paw orientation at 1 mm past the pellet (linear mixed model: effect of laser: $t(607)$ = 0.88, p=0.38; interaction between laser and session: $t(607)$ = −0.55, p=0.58). SNc illumination in EYFP-injected rats had no effect on paw orientation at 1 mm past the pellet (linear mixed model: effect of laser: $t(607)$ = −0.51, p=0.61; interaction between laser and session: $t(607)$ = 0.31, p=0.76). Gray points indicate data from individual rats. * indicates p<0.05 for either the laser or laser-session interaction terms in panels C and F. Aperture and orientation as a function of $z_{digit2}$ for successful and failed reaches are shown in *Figure 9—figure supplements 2,3*, respectively. Aperture and orientation as a function of $z_{digit2}$ are shown for each rat individually in *Figure 9—figure supplements 4,5*, respectively.

The online version of this article includes the following source data and figure supplement(s) for figure 9:

**Source data 1.** A .mat file containing aperture (mean_aperture_traj) and paw orientation (mean_orientation_traj) data along reach trajectories (see Materials and methods for details of calculations) for 22 testing sessions ('retraining', 'laser on', and 'occluded').

**Figure supplement 1.** Dopamine manipulations between reaches do not affect reach-to-grasp coordination.

**Figure supplement 1—source data 1.** A .mat file containing aperture (mean_aperture_traj) and paw orientation (mean_orientation_traj) data along reach trajectories (see Materials and methods for details of calculations) for 22 testing sessions ('retraining', 'laser on', and 'occluded').

**Figure supplement 2.** Mean aperture as a function of paw advancement separated by reach success or failure.

**Figure supplement 2—source data 1.** A .mat file containing aperture (mean_aperture_traj) data along reach trajectories (see Materials and methods for details of calculations) separated according to trial outcome.

**Figure supplement 2—source data 2.** Statistics.

**Figure supplement 3.** Mean paw orientation as a function of paw advancement separated by reach success or failure.

**Figure supplement 3—source data 1.** A .mat file containing paw orientation (mean_orientation_traj) data along reach trajectories (see Materials and methods for details of calculations) separated according to trial outcome.

**Figure supplement 3—source data 2.** Statistics.

**Figure supplement 4.** Mean aperture as a function of paw advancement for each rat.

**Figure supplement 4—source data 1.** A .mat file containing aperture (mean_aperture_traj) data along reach trajectories (see Materials and methods for details of calculations) for 22 testing sessions ('retraining', 'laser on', and 'occluded').

**Figure supplement 5.** Mean paw orientation as a function of paw advancement for each rat.

**Figure supplement 5—source data 1.** A .mat file containing paw orientation (mean_orientation_traj) data along reach trajectories (see Materials and methods for details of calculations) for 22 testing sessions ('retraining', 'laser on', and 'occluded').

The abrupt transitions between aberrant and baseline reach kinematics are reminiscent of 'on/off' motor fluctuations observed in people with PD. With disease progression and prolonged treatment, patients often display sudden transitions between severe bradykinesia, good motor control, and levodopa-induced dyskinesias (*Chou et al., 2018*). Disease duration, degree of dopamine loss, and magnitude of treatment-related dopamine fluctuations are correlated (*Abercrombie et al., 1990*; *de la Fuente-Fernández et al., 2004*). The root causes of motor fluctuations are therefore difficult to identify, though it is suggested that 'on-offs' in PD may share mechanisms with drug sensitization (*Calabresi et al., 2008*). Our results indicate that large, temporally specific dopamine fluctuations are sufficient to cause dramatic dopamine-dependent changes in movement kinematics, even in otherwise healthy subjects. This suggests that large swings in striatal dopamine are sufficient to generate motor fluctuations, independent of the degree of dopamine denervation.

The motor effects of dopamine neuron stimulation also depended on behavioral context. Dopamine neuron stimulation had almost no effect on stimulation-naive rats in a clear cylinder. Rats engaged in skilled reaching during dopamine neuron stimulation continued to engage in the task,

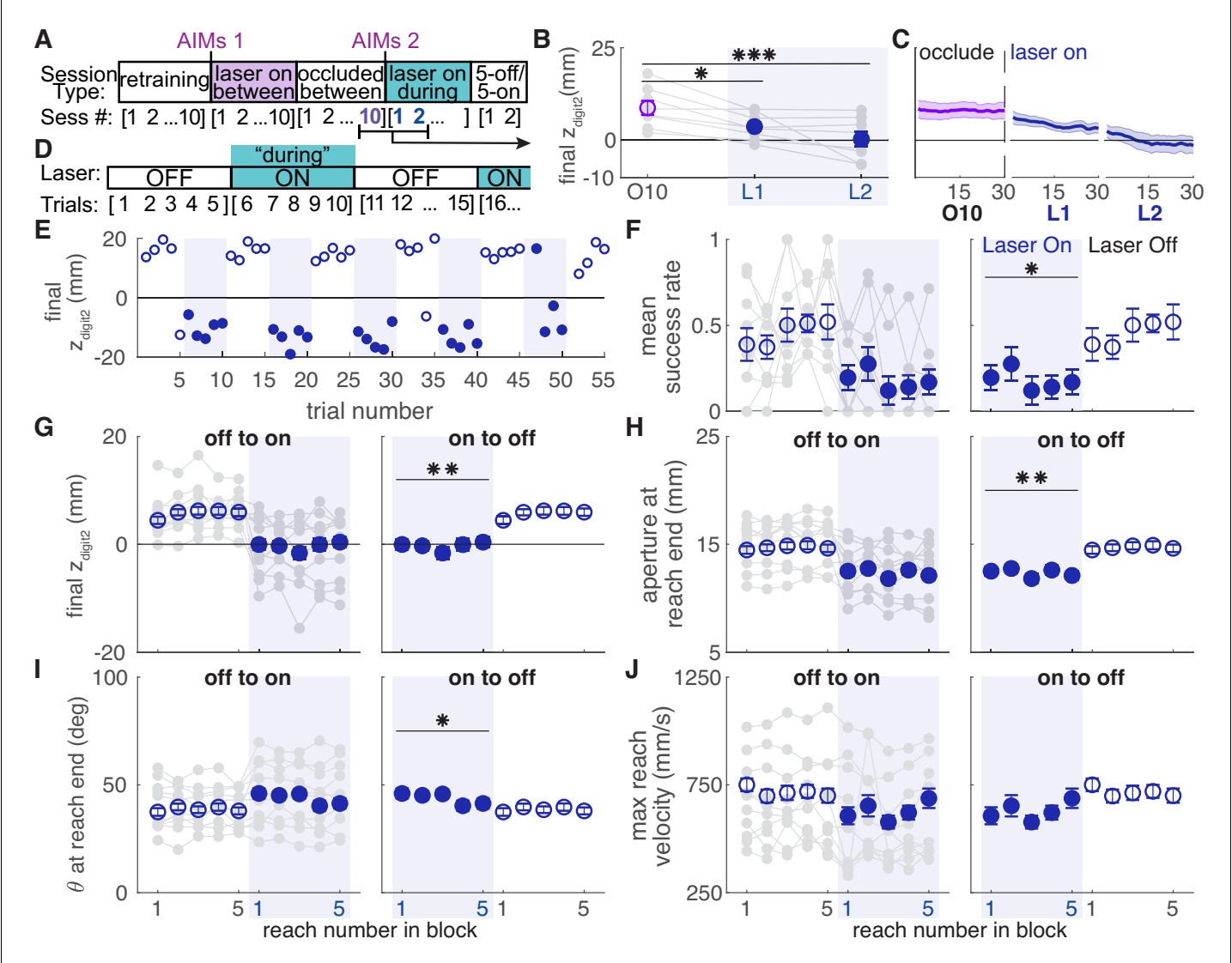

**Figure 10.** Dopamine neuron stimulation induces distinct reach-to-grasp kinematics that depend on current dopamine stimulation. (A) Experimental timeline for 'ChR2 between' rats. After 10 sessions of testing with the laser occluded, 'ChR2 between' rats underwent two to eight sessions (average: 3.17 ± 0.98 sessions) with 'during reach' stimulation. 'AIMs 1' and 'AIMs 2' indicate the timing of abnormal involuntary movement testing. Once reaching movements were impaired (Panels B-C and *Figure 10—figure supplements 1,2*), rats underwent testing with laser off and on for blocks of five trials (Panels D-J). (B) Average maximum reach extent on the last day of testing with the laser occluded between reaches (O10) and the first 2 days of testing with the laser on during reaches (L1 and L2). Maximum reach extent decreased significantly between O10 and L1 (linear mixed model: $t(14)$ = −2.58, p=0.02) and O10 and L2 (linear mixed model: $t(14)$ = −4.33, p=6.92×10$^{-4}$). (C) Moving average of maximum reach extent for last day of testing with the laser occluded between reaches and the first 2 days of testing with the laser on during reaches. Additional kinematic measures (aperture, orientation, reach velocity) are shown in *Figure 10—figure supplement 1*. Individual rat data are shown in *Figure 10—figure supplement 2*. (D) Schematic of alternating stimulation test sessions. (E) Example session from one rat with maximum reach extent plotted for every trial. Some blocks have fewer than five trials if the rat did not reach for the pellet after breaking the IR beam. (F) Average first attempt success rate during 'laser off' and 'laser on' blocks. Data are repeated to show 'off to on' and 'on to off' transitions in panels F-J. Gray lines show individual rat data. Linear mixed model: effect of laser: $t(78)$ = −0.50, p=0.62; interaction between laser and trial within block: $t(78)$ = −2.35, p=0.02. (G) Average maximum reach extent during 'laser off' and 'laser on' blocks. Linear mixed model: effect of laser: $t(78)$ = 2.70, p=8.47×10$^{-3}$; interaction between laser and trial within block: $t(78)$ = 1.32, p=0.19. (H) Average aperture at reach end across 'laser off' and 'laser on' blocks. Linear mixed model: effect of laser: $t(78)$ = −2.83, p=5.92×10$^{-3}$; interaction between laser and trial within block: $t(78)$ = −0.79, p=0.43. (I) Average paw orientation at reach end across 'laser off' and 'laser on' blocks. Linear mixed model: effect of laser: $t(78)$ = 2.57, p=0.01; interaction between laser and trial within block: $t(78)$ = −0.34, p=0.73. (J) Average maximum reach velocity across 'laser off' and 'laser on' blocks. Linear mixed model: effect of laser: $t(78)$ = −1.24, p=0.22; interaction between laser and trial within block: $t(78)$ = 0.01, p=0.99. * indicates p<0.05 for effect of laser in panel F. ** indicates p<0.01 for effect of laser in panel D.

The online version of this article includes the following source data and figure supplement(s) for figure 10:

*Figure 10 continued on next page*

*Figure 10 continued*

**Source data 1.** A .mat file containing zdigit2 endpoint (mean_dig2_endPt) data for 22 testing sessions ('retraining', 'laser on', and 'occluded').

**Source data 2.** A .mat file containing zdigt2 endpoint (mean_dig2_endPt) data for 'during reach' sessions in originally 'ChR2 Between' rats.

**Source data 3.** A .mat file containing digit2 endpoint (digEnd) averages across a moving block of 10 trials for 22 testing sessions ('retraining', 'laser on', and 'occluded').

**Source data 4.** A .mat file containing digit2 endpoint (digEnd) averages across a moving block of 10 trials for 'during reach' sessions in originally 'ChR2 Between' rats.

**Source data 5.** A .mat file containing '5off/5on' data.

**Figure supplement 1.** Additional kinematic measures for 'during reach' stimulation testing in 'ChR2 between' rats.

**Figure supplement 1—source data 1.** A .mat file containing aperture (mean_end_aperture), paw orientation (mean_end_orientations), and velocity (mean_pd_v) data for 22 testing sessions ('retraining', 'laser on', and 'occluded').

**Figure supplement 1—source data 2.** A .mat file containing aperture (mean_end_aperture), paw orientation (mean_end_orientations), and velocity (mean_pd_v) data for 'during reach' sessions in originally 'ChR2 Between' rats.

**Figure supplement 1—source data 3.** A .mat file containing aperture, paw orientation, and velocity averages across a moving block of 10 trials for 22 testing sessions ('retraining', 'laser on', and 'occluded').

**Figure supplement 1—source data 4.** A .mat file containing aperture, paw orientation, and velocity averages across a moving block of 10 trials for 'during reach' sessions in originally 'ChR2 Between' rats.

**Figure supplement 2.** Individual rat data for moving average of maximum reach extent, aperture at reach end, paw orientation at reach end, and maximum reach velocity within individual sessions.

**Figure supplement 2—source data 1.** A .mat file containing digit2 endpoint, aperture, paw orientation, and velocity averages across a moving block of 10 trials for 22 testing sessions ('retraining', 'laser on', and 'occluded').

**Figure supplement 2—source data 2.** A .mat file containing digit2 endpoint, aperture, paw orientation, and velocity averages across a moving block of 10 trials for 'during reach' sessions in originally 'ChR2 Between' rats.

with few abnormal involuntary movements during reaching (*Figure 11*). However, the same stimulation parameters delivered to previously-stimulated rats in clear cylinders induced markedly abnormal limb and body movements (*Figure 11* and *Video 6*). Thus, skilled reaching sensitization may have generalized to the cylinder. However, it is unclear if enhanced AIMs required stimulation in the reaching chamber, or if stimulation during the initial AIMs test was sufficient (a single amphetamine dose can induce sensitization given a sufficiently long drug hiatus, *Valjent et al., 2010*; *Vanderschuren et al., 1999*). In either case, this is consistent with the idea that dopamine regulates the 'vigor' of movements selected based on the current behavioral context (*Yttri and Dudman, 2016*). That is, in the reaching chamber, rats approach the reaching slot to perform a (dopamine-modified) reach because that is the appropriate action in that context. Conversely, with no specific goal-directed actions suggested by the cylinder context, dopamine equally invigorates many potential movements. This leads to seemingly random abnormal involuntary movements (*Bastide et al., 2015*). Interestingly, the severity of experimental levodopa-induced dyskinesias depends on behavioral context (*Lane et al., 2011*). This context dependence of dopaminergic effects on motor control has parallels in clinical phenomenology: people with PD often can perform goal-directed movements despite the presence of significant levodopa-induced dyskinesias.

There are several limitations of this study. First, we did not record from dopamine neurons or measure dopamine release during optogenetic manipulations. It is therefore not clear how striatal dopamine levels were altered relative to normal reach-related dopamine dynamics, or if repeated stimulation changed spontaneous or

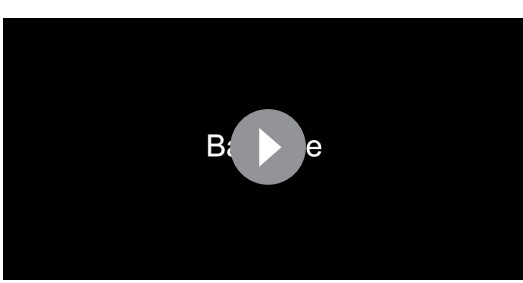

**Video 5.** Sample reaches from a rat that received 'during reach' stimulation in alternating trial blocks demonstrating that kinematic changes induced by dopamine neuron stimulation are enduring. Reach 1 – at baseline, the rat extends its paw past the pellet to grasp it. Reach 2 – after several reaches with stimulation, the second digit extends just to the pellet, which is knocked off the pedestal. Reach 3 – after more reaches with stimulation, the reach comes far short of the pellet. Reach 4 – with stimulation off, reach kinematics return to baseline. Reach 5 – on the next reach, stimulation is reinstated and kinematics are markedly abnormal.
https://elifesciences.org/articles/61591#video5

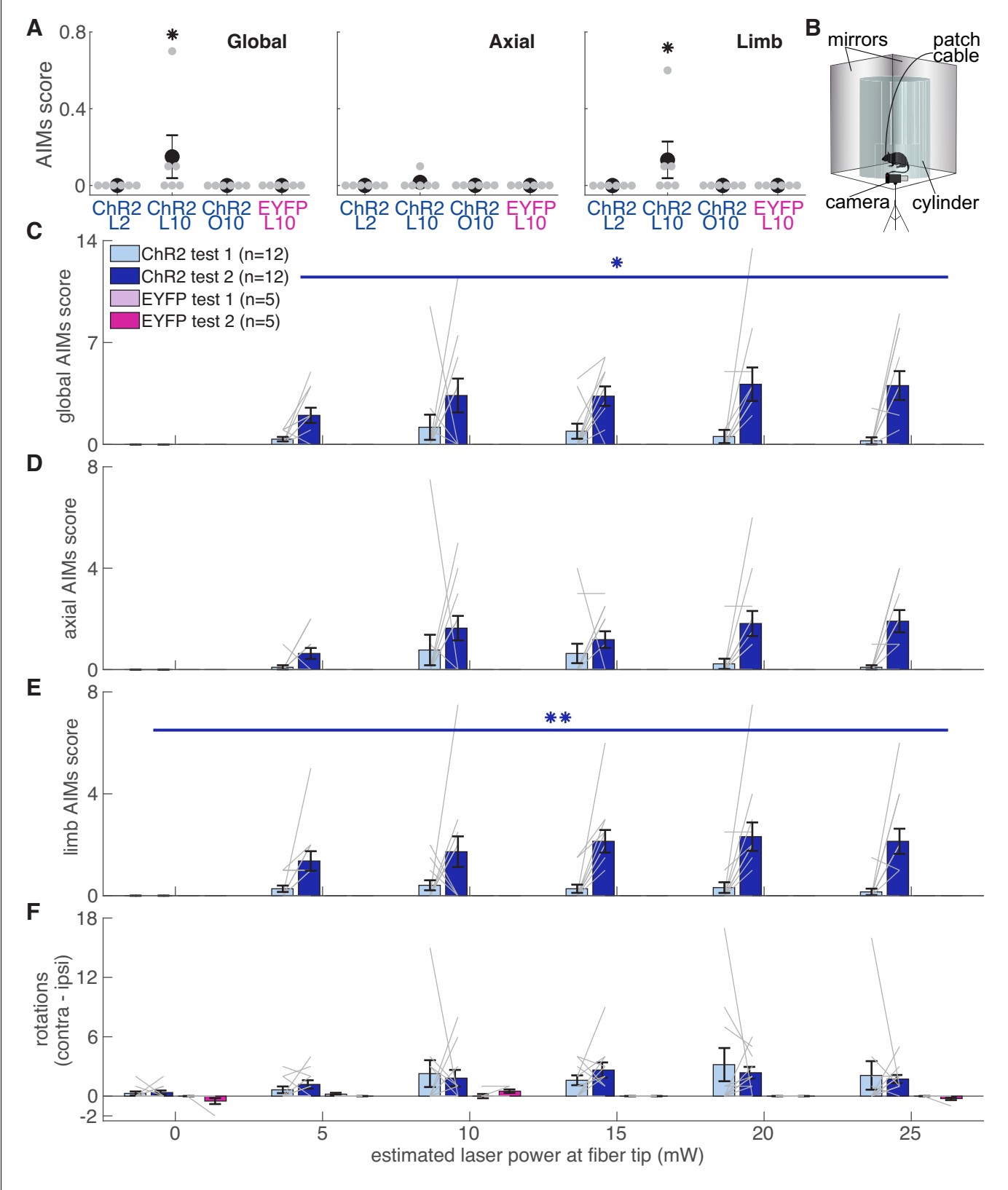

**Figure 11.** Dopamine neuron stimulation induces context- and history-dependent abnormal involuntary movements. (**A**) Average AIMs scores during reaches for 'ChR2 during' rats on days 2 and 10 of 'laser on' sessions, and day 10 of 'occlusion' sessions. Also, AIMs scores during reaches for 'EYFP'
*Figure 11 continued on next page*

*Figure 11 continued*

rats on 'laser on' day 10. Global (Kruskal-Wallis: $\chi^2(3)$=9.31, p=2.54×10$^{-2}$) and limb ($\chi^2(3)$=9.31, p=2.54×10$^{-2}$) AIM scores were higher in 'laser on' session 10 for ChR2 rats. Axial AIM scores did not differ between groups ($\chi^2(3)$=2.83, p=0.42). * indicates p<0.05. (B) Experimental set-up for AIMs test. (C) Average global AIMs scores vs. estimated power at the fiber tip. Global AIMs increased with increasing laser power and from test day 1 to 2 in ChR2-injected rats (linear mixed model: interaction between test number and laser power: $t(164)$ = 2.57, p=0.01). EYFP-injected rats did not display AIMs (linear mixed model: interaction between test number and laser power: $t(164)$ = 0.00, p=1.00). ChR2-injected rats' global AIMs scores did not differ significantly from EYFP-injected rats' global AIMs scores on test day 1 (Contrast tests: 5 mW: $t(164)$ = 1.57, p=0.12; 10 mW: $t(164)$ = 1.71, p=0.09; 15 mW: $t(164)$ = 1.45, p=0.15; 20 mW: $t(164)$ = 1.17, p=0.24; 25 mW: $t(164)$ = 0.98, p=0.33). Gray lines represent data from individual rats. Error bars represent s.e.m. across animals. (D) Average axial AIMs scores. ChR2: linear mixed model: interaction between test number and laser power: $t(165)$ = 1.91, p=0.06. EYFP: $t(165)$ = 0.00, p=1.00. ChR2-injected rats' axial AIMs scores did not differ significantly from EYFP-injected rats' axial AIMs scores on test day 1 (Contrast tests: 5 mW: $t(165)$ = 1.36, p=0.18; 10 mW: $t(165)$ = 1.46, p=0.15; 15 mW: $t(165)$ = 1.22, p=0.23; 20 mW: $t(165)$ = 0.97, p=0.33; 25 mW: $t(165)$ = 0.80, p=0.43). (E) Average limb AIMs scores. A linear mixed-effects model found a significant interaction between test number and laser power in ChR2-injected rats: $t(164)$ = 2.81, p=5.51×10$^{-3}$. EYFP-injected rats did not display limb AIMs: $t(164)$ = 0.00, p=1.00. ChR2-injected rats' limb AIMs scores did not differ significantly from EYFP-injected rats' limb AIMs scores on test day 1 (Contrast tests: 5 mW: $t(164)$ = 1.42, p=0.16; 10 mW: $t(164)$ = 1.55, p=0.12; 15 mW: $t(164)$ = 1.33, p=0.19; 20 mW: $t(164)$ = 1.08, p=0.28; 25 mW: $t(164)$ = 0.91, p=0.37). (F) Difference between average number of contralateral and ipsilateral (relative to hemisphere implanted with optical fiber) rotations. A positive score indicates a bias toward contralateral spins and a negative score indicates a bias towards ipsilateral spins. ChR2-injected rats did not increase the number of contralateral spins between test 1 and test 2, nor did laser power affect rotational behavior. Linear mixed model: interaction between test number and laser power: $t(164)$ = −0.39, p=0.69. EYFP-injected rats did not show a bias in either direction with laser stimulation: $t(164)$ = 0.10, p=0.92. (*p<0.05, **p<0.01 for ChR2-injected rats). ChR2 rats had a stronger bias toward contralateral spins compared to EYFP rats at 5 mW (contrast tests: $t(164)$ = 2.57, p=0.01), 10 ($t(164)$ = 3.44, p=7.37×10$^{-4}$), 15 ($t(164)$ = 3.38, p=9.19×10$^{-4}$), 20 mW ($t(164)$ = 3.05, p=2.69×10$^{-3}$), and 25 mW ($t(164)$ = 2.77, p=6.29×10$^{-3}$) on test day 1. ChR2 rats in panels C-F include rats from 'ChR2 during' (n = 5) and 'ChR2 between' (n = 7) groups. *Figure 11—figure supplement 1* shows the relationship between AIMs scores in the cylinder and reach success rates for individual rats.

The online version of this article includes the following source data and figure supplement(s) for figure 11:

**Source data 1.** A .mat file containing number of rotations to the left (leftSpin) and right (rightSpin), axial AIMs scores (axialAmplitude and axialBasis), and limb AIMs scores (limbAmplitude and limbBasic) from AIMs tests 1 and 2.

**Source data 2.** A .mat file containing limb and axial amplitude scores for randomly selected skilled reaching trials.

**Figure supplement 1.** Dyskinesias in the second AIMs test do not consistently predict changes in reach success rate.

**Figure supplement 1—source data 1.** A.mat file containing axial AIMs scores (axialAmplitude and axialBasis) and limb AIMs scores (limbAmplitude and limbBasic) from AIMs tests 1 and 2.

**Figure supplement 1—source data 2.** A .mat file containing first reach success rate data (firstReachSuccess) for 22 testing sessions ('retraining', 'laser on', and 'occluded').

**Figure supplement 1—source data 3.** A .mat file containing containing first reach success rate data (firstReachSuccess) for 'during reach' sessions in originally 'ChR2 Between' rats.

optically evoked dopamine release (*Saunders et al., 2018*). Given the relatively high optical stimulation power (20 mW at the fiber tip) and frequency (20 Hz) used, we suspect that we induced supraphysiologic dopamine release (*Patriarchi et al., 2018*, but see *Hamid et al., 2016*). Nonetheless, supra/infraphysiologic manipulations (e.g. lesion studies) can provide important insights into normal function. Furthermore, supraphysiologic dopamine fluctuations are relevant to pathologic states like PD, in which striatal dopamine can transition over minutes to hours between very low and high levels (*Abercrombie et al., 1990*; *de la Fuente-Fernández et al., 2004*). Second, we stimulated over SNc. It is therefore unclear how ventral tegmental area (VTA) stimulation would influence skilled reaching, and whether stimulating specific nigral projection fields (e.g. striatal subregions or motor cortex, *Guo et al., 2015*, *Hosp et al., 2011*, *Zhang et al., 2017*) would differentially affect reach kinematics. Finally, while we found that dopamine neuron manipulations during, but not between, reaches affected reach kinematics, the timing of when dopamine manipulations exert their effects could be parsed more precisely. Our 'during reach' timing covered approach to the pellet, the reach itself, and immediately after the grasp during pellet consumption. 'Phasic' dopamine release at the time of an unpredicted reward is believed to be responsible for reinforcing actions of dopamine, while 'tonic' dopamine levels are suggested to regulate motivation/vigor (*Niv et al., 2007*). Activation of different terminal fields at different times with respect to behavior, as well as continuous monitoring of dopamine release (*Patriarchi et al., 2018*), may identify distinct roles for tonic and phasic dopamine in dexterous skill.

In summary, temporally specific dopamine signals cause gradual changes in dexterous skill performance separable from pure 'vigor' effects. These changes are durable, and expressed in a

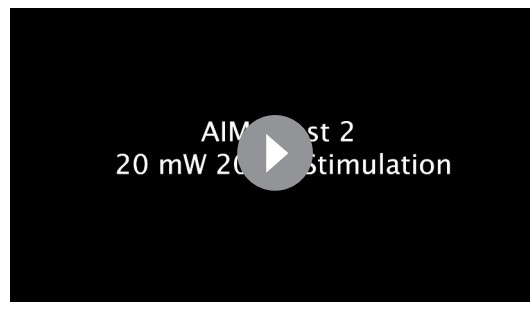

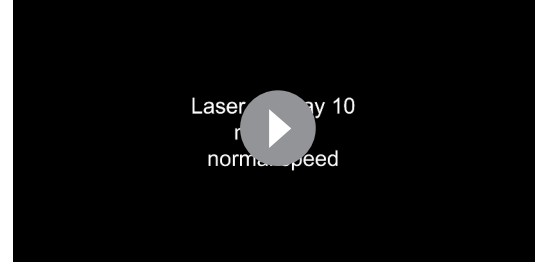

**Video 6.** Context-dependent AIMs. ChR2-injected rat showing AIMs (axial and limb dyskinesias) with dopamine neuron stimulation during the second day of AIMs testing. The second segment shows a reach in which the same rat does not show AIMs when receiving the same stimulation parameters (estimated 20 mW at the fiber tip, 20 Hz) during a reach.
https://elifesciences.org/articles/61591#video6

**Video 7.** AIMs during reaching in a 'ChR2 during' rat during 'laser on' session 10 (same rat as in *Video 6*). The first segment shows an example of a non-dyskinetic reach. While the reach is hypometric, the motion is smooth. The last segment shows a reach during the same session in which dyskinesias interfered with the reach (limb amplitude score = 1). The red dot indicates moments when the rat is showing AIMs (paw is oscillating from left to right instead of moving towards or away from the pellet). Videos are shown at normal and 20% speed.
https://elifesciences.org/articles/61591#video7

dopamine-dependent manner on a reach-by-reach basis. This phenomenon has clinical analogy with rapid motor fluctuations in PD patients. It may, therefore, serve as a useful paradigm in which to study the underlying neurobiology of motor fluctuations in PD, as well as address fundamental questions regarding how dopamine and basal ganglia circuits regulate skilled movements.

## Materials and methods

**Key resources table**

| Reagent type (species) or resource | Designation | Source or reference | Identifiers | Additional information |
|---|---|---|---|---|
| Strain, strain background (*TH-Cre Long-Evans rats, male and female*) | LE-Tg(TH-Cre)3.1Deis | Rat Resource and Research Center | RRRC#: 00659; RRID:RGD_10401201 | |
| Antibody | anti-GFP (Mouse monoclonal) | Millipore Sigma | Cat. #: MAB3580; RRID:AB_94936 | (1:1500) |
| Antibody | anti-TH (Rabbit polyclonal) | Millipore Sigma | Cat. #: AB152; RRID:AB_390204 | (1:2000) |
| Antibody | Alexa Fluor 488 donkey anti-mouse IgG (Donkey polyclonal) | Thermo Fisher | Cat. #: A-21202; RRID:AB_141607 | (1:500) |
| Antibody | Alexa Fluor 555 donkey anti-rabbit IgG (Donkey polyclonal) | Thermo Fisher | Cat. #: A-31572; RRID:AB_162543 | (1:500) |
| Recombinant DNA reagent | AAV5-EF1α-DIO-hChR2 (H134R)-EFYP | UNC Vector Core | | |
| Recombinant DNA reagent | AAV5-EF1α-DIO-eArch3.0-EYFP | UNC Vector Core | | |
| Recombinant DNA reagent | AAV-EF1α-DIO-EYFP | UNC Vector Core | | |
| Software, algorithm | Matlab | Mathworks | RRID:SCR_001622 | |
| Software, algorithm | RStudio | RStudio PBC | RRID:SCR_000432 | |
| Other | ProLong Diamond | Invitrogen | Cat. #: P36961 | |

## Rats

All animal procedures were approved by the University of Michigan Institutional Animal Care and Use Committee. Numbers of rats included in each experimental group and analysis are indicated in figure legends and the main text. For each experiment (e.g. ChR2 'During Reach' vs EYFP controls), rats were randomly allocated to active opsin or EYFP control groups. Male (n = 23) and female (n = 15) TH-Cre⁺ rats were housed in groups of 2–3 on a reverse light/dark cycle prior to optical fiber implantation. Following surgery, rats were housed individually to protect the implant. All testing was carried out during the dark phase. Food restriction was imposed on all animals during the training and testing periods for no more than 6 days in a row such that rats' weights were kept ~85–90% of their free-feeding weight. Water was available ad libitum in their home cages. Eight rats were excluded from the analysis due to either poor opsin expression or misplaced optical fibers (number of rats excluded: Group 1: n = 1; Group 2: n = 3; Group 3: n = 3; Group 4: n = 0; Group 5: n = 1). Judgment on whether to include subjects was made by investigators blinded to experimental groups and outcomes.

## Stereotaxic surgeries

Before pre-training for skilled reaching, rats were anesthetized with isoflurane (5% induction and 2–3% maintenance) and bilaterally injected in the SNc (M-L ±1.8 mm; A-P −5.2 mm, −6.2 mm; D-V −7.0 mm, −8.0 mm) with AAV-EF1α-DIO-hChR2(H134R)-EFYP, AAV-EF1α-DIO-eArch3.0-EYFP, or AAV-EF1α-DIO-EYFP (UNC vector core). 1 µl of virus (titer: 3.4–4.2 × 10¹² vg/ml) was injected per site (4 µl total per hemisphere) at a rate of 0.1 µl/min. After reaching stable performance on the skilled reaching task, optical fibers (multimode 200 µm core, 0.39 NA, Thor Labs FT200EMT) embedded in stainless steel ferrules (2.5 mm outer diameter, 230 µm bore size, Thor Labs #SF230-10) were implanted above SNc contralateral to the rat's preferred reaching paw (M-L ±2.4 mm, A-P −5.3 mm, D-V −7.0 mm). Optical fibers were calibrated before implantation to determine optical power at the fiber tip as a function of laser output power, which was continuously monitored during experiments by 'picking off' 10% of the laser output with a beamsplitter. Rats recovered for at least 7 days after surgical procedures before beginning behavioral training or testing.

## Skilled reaching
### Automated reaching system

Training and testing were carried out in custom-built skilled reaching chambers housed within sound-proof, ventilated cabinets (Figure 1D, Bova et al., 2019; Ellens et al., 2016). Infrared sensors (HoneyWell, Morriston, NJ) were aligned so that the beam was directed through the back of the chamber. A reaching slot (1.1 × 7 cm) was cut into the front panel of the chamber 3.5 cm from the floor. One mirror was placed on either side of the front of the reaching chamber and angled to allow side views of the paw during reaches. A linear actuator with three-position digital control (Creative Werks Inc, Des Moines, IA) was connected to an acrylic pellet delivery rod and mounted in a custom frame below the support box. The pellet delivery rod extended through a funnel mounted to the top of the frame. Before each session, the actuator was positioned so that the delivery rod was aligned with the right or left edge of the slot according to each rat's paw preference 15 mm from the front of the reaching slot.

Videos were recorded at 300 frames-per-second and 2400 × 1024 pixels by a high-definition color digital camera (acA2000-340kc, Basler, Ahrensburg, Germany) mounted in front of the reaching slot. A camera-link field-programmable gate array (FPGA) frame-grabber card (PCIe 1473R, National Instruments, Austin, TX) acquired the images, and an FPGA data acquisition (DAQ) task control card (NI PCIe 7841R) provided an interface with the behavior chamber and optogenetic system. The real-time FPGA card detected pixel intensity changes within a 'region of interest' in front of the reaching slot visible in the side mirror views (Figure 1D), allowing videos of the reaching event ('video trigger') to be captured. 300 frames pre-trigger and 1000 frames post-trigger were saved. A second camcorder was placed above the reaching chamber to record the entire session at 60 frames-per-second (HC-V110, Panasonic).

 

### Trial performance

Custom LabVIEW software controls the experiment (*Bova et al., 2019*; *Ellens et al., 2016*). Each training session begins with the pellet delivery rod in the 'ready' position - halfway between the bottom of the reaching chamber and the reaching slot. When the rat breaks the IR beam at the back of the chamber, the pellet delivery rod rises to the bottom of the reaching slot. When the reaching paw passes the front plane of the chamber into the 'region of interest' and surpasses the minimum threshold of pixel intensity, video acquisition is triggered, time-stamped, and labeled with the trial number. Two seconds after the video is triggered, the pellet delivery arm lowers into the pellet funnel to pick up a new pellet and then resets to the 'ready' position, allowing the rat to initiate a new trial.

### Pre-training

'Pre-training' consists of familiarizing the rats with the reaching chamber, evaluating them for paw-preference, training them to reach for the linear actuator, and training them to request a pellet by moving to the back of the chamber. A week before pre-training, rats were placed on food restriction and introduced to the sucrose reward pellets in their home cages. On day 1 of pre-training, piles of five pellets each were placed in the front and rear of the skilled reaching chamber to encourage exploration of the entire chamber. Once rats ate these pellets, they were evaluated for paw-preference.

Rats were allowed to eat three pellets (held in forceps through the reaching slot) with their tongues. The experimenter then began to pull the pellet away from the rat so that it could not be obtained by licking. Therefore, the rat was forced to reach with its paw to retrieve the pellet. Paw preference was assigned to the paw used for the majority of the first eleven reaches. Once paw preference was determined, animals were trained to reach for the pellet delivery rod. As the rat reached, the experimenter pulled the forceps back so that the rat's paw would extend to a pellet on the delivery rod. Once rats reached for the delivery rod 10 times without being baited by the experimenter, they began training to request pellets.

Rats began training in the center of the chamber with the pellet delivery rod set to the 'ready' position. The experimenter placed a pellet in the rear of the chamber to bait the rat to break the rear IR beam, causing the delivery rod to rise so that the rat could move to the front and reach for the pellet. This was repeated until the rat began to quickly move to the front of the chamber to reach for the pellet after breaking the IR beam. At this point, the experimenter would stop baiting the rat to the rear of the chamber. Pre-training was complete once the rat requested a pellet and then immediately moved to the front to reach for the pellet 10 times.

### Training

After pre-training, rats began 30-min training sessions with the automated system. Rats were trained for 6 days per week until they reached stable performance (minimum of 35 reaches and a steady success rate above 40% over three sessions). Once behavioral criteria were met, rats were implanted with optical fibers.

## Optogenetics

Before testing with optogenetic interventions, rats were re-trained for 10 days while tethered to the patch cable without light delivery. This allowed rats to return to stable performance after surgery and adapt to the tether. During the 10 days of testing with optogenetic interventions, light was delivered on every trial at one of two different times. For 'during reach' stimulation, the laser turned on when the rat broke the IR beam at the back of the chamber and remained on until 3 s after the video trigger event. For 'between reach' stimulation, light was delivered beginning 4 s after the video trigger and remained on for 5 s (*Figure 1D*). The duration of 'between reach' stimulation was approximately matched to the median duration of 'during reach' stimulation (*Figure 1—figure supplement 1*). The duration of 'during reach' stimulation was assessed by custom MATLAB code (*Bova et al., 2020*), which scanned videos recorded from above the reaching chamber frame-by-frame for blue or green laser light, and identified the beginning and ends of 'laser on' bouts. An experimenter evaluated the output for accuracy and corrected any errors (e.g. rat's head went out of frame and one laser bout was incorrectly split into two bouts). For ChR2- and EYFP-injected rats,

473 nm laser light (Opto Engine DPSS laser) was delivered at 20 Hz and an estimated 20 mW at the fiber tip based on pre-implantation measurements using a calibrated photodiode (Thorlabs S121C connected to Thorlabs PM100D Power Meter). The laser was on continuously, with 20 Hz stimulation achieved using an optical chopper (Thorlabs MC1F10HP) to eliminate transient power fluctuations as the laser is turned off and on. For Arch-injected rats, 532 nm laser light (Opto Engine DPSS laser) was delivered continuously at an estimated 20 mW at the fiber tip.

Following optogenetic testing, rats were tested for another 10 days with the patch cable attached to the implanted fiber and the laser activated. However, the patch cable-implanted fiber junction was physically occluded by inserting a piece of dense foam within the connector that holds the patch cable and optical fiber. Full occlusion of the laser was checked before each session by measuring light output at the fiber tip using a calibrated photodiode (Thorlabs S121C connected to Thorlabs PM100D Power Meter). In this way, all sensory cues were identical (e.g. visible light, optical shutter sounds) but light could not penetrate into the brain. The timing of light delivery was identical to that used during testing with optogenetic interventions.

Following the last 'occlusion' session, ChR2 Between rats (see *Figures 1* and *10A*) were tested for an additional 2–8 days with the laser on during reaches, until reaches were impaired. Rats then performed an additional one or two 30-min sessions during which the laser alternated every five trials between being off and on during reaches.

## Analysis of skilled reaching data

Analyses were performed using custom-written scripts and functions in MATLAB 2019a (MathWorks, *Bova et al., 2020*).

## Number of trials and success rate

Reach outcome was scored by visual inspection as follows: 0 – no pellet presented or other mechanical failure ('no pellet'); 1 – first trial success (obtained pellet on initial limb advance; 'first success'); 2 – success (obtained pellet, but not on first attempt; 'multi try success'); 3 – forelimb advanced, pellet was grasped then dropped in the box ('drop in box'); 4 – forelimb advance, but the pellet was knocked off the shelf ('pellet knocked off'); 5 – pellet was obtained using its tongue ('used tongue'); 6 – the rat approached the slot but retreated without advancing its forelimb or the video triggered without a reach ('trigger error'); 7 – the rat reached, but the pellet remained on the shelf ('pellet remained'); 8 – the rat used its ipsilateral (in reference to hemisphere with optical fiber implant) paw to reach ('ipsilateral paw'); 9 – laser fired at the wrong time ('laser error'); or 10 – used preferred paw after obtaining or moving pellet with tongue ('tongue and paw'). An additional outcome evaluated by kinematic analysis was defined as videos which began with the rat's paw through the slot (i.e. the video triggered late; 'paw through slot'). Outcome percent was calculated by dividing the number of trials of each outcome by the total number of trials per session. For comparisons of kinematics between successful and failed reaches, successes were defined as 'first success' (1) and failed reaches were defined as 'pellet knocked off' (4) or 'pellet remained' (7).

First reach success was calculated for each session by dividing the total number of scores of 1 by the total number of trials (sum of scores of 1, 2, 3, 4, and 7). 'Any reach' success rate was calculated by dividing the sum of scores 1 and 2 by the total number of trials. For both number of trials per session and first reach success rate, a baseline score was calculated for each rat by averaging the scores of the last two retraining sessions (*Figure 2A,D*). Number of trials and success rates for each session within 'laser on' and 'occlusion' sessions were normalized by dividing the score for that session by the averaged baseline score (*Figures 2B–C, E–F,3A–B,4A–D*).

To assess how success rate changed within individual sessions, a moving average was calculated as the fraction of '1' scores in a moving block of 10 reaches. For averages within a group, the last data point for each individual was carried forward to the maximum number of reaches for any rat in that session. This avoided sudden changes in the average caused by dropout (*Figure 2G* and *Figure 2—figure supplement 1*, *Figure 3C* and *Figure 3—figure supplement 1*, *Figure 4E* and *Figure 4—figure supplements 1–2*).

### Three-dimensional reconstruction of reach trajectories

Bodyparts/objects identified in the direct and mirror views were triangulated to three-dimensional points using custom MATLAB software (*Bova et al., 2019*). Prior to each session, several images of a cube with checkerboards (4 × 4 mm squares) on its sides were taken so that the checkerboards were visible in the direct and mirror views. These images were used to determine the essential matrix relating the direct and mirror views, which was used to determine how the real camera and 'virtual' camera behind the mirror were translated and rotated with respect to each other (*Hartley and Zisserman, 2003*). By assuming a three-dimensional coordinate system centered at the camera lens with the z-axis perpendicular to the lens surface, camera matrices were derived for the real and virtual cameras. These matrices were used to triangulate matching points in the camera and mirror views using the MATLAB triangulate function in the Computer Vision toolbox. Three-dimensional points with large reprojection errors were excluded from the analysis, which could happen if an object was identified accurately in one view but misidentified in the other. The coordinate system was set with the pellet at the origin, positive x to the right of the pellet, positive y below the pellet, and positive z on the other side of the pellet from the reaching chamber (*Figure 5*).

### Processing reach kinematics

To place reach kinematics in a common reference frame, the pellet location prior to reaching was identified and set as the origin. For left-pawed reaches, x-coordinates were negated to allow direct comparison with right-pawed reaches. The initial reach on each trial was identified by finding the first frame in which digits were visible outside the box, and then looking backwards in time until the paw started moving forward. The end of a reach was defined as the frame at which the tip of the second digit began to retract ('maximum reach extent', $z_{digit2}$, *Figure 5*); multiple reaches could be counted in a single trial. 'Aperture' was calculated as the Euclidean distance between the tips of the 1st and 4th digits (in frames for which both were visible or could be estimated based on epipolar geometry). 'Orientation' was calculated as the angle between a line connecting the 1st and 4th digits and a horizontal line (for left-pawed rats, orientation was calculated using the negated x-values to compare with right-pawed rats). Paw velocity was calculated as the Euclidean distance between the dorsum of the reaching paw in consecutive frames divided by the inter-frame interval (1/300 s).

To separate the influences of changes in paw transport from changes in grasp kinematics on reaching proficiency, we examined success rates across experimental groups in trials matched for reach extent (*Figure 6—figure supplement 8* and *Figure 7—figure supplement 8*). We stratified trials by their maximum $z_{digit2}$ extent in 1-mm-wide bins beginning at $z_{digit2} = 0$, and compared reach success, digit aperture, and paw orientation within each stratum.

### Within-session kinematics

To assess how reach kinematics (i.e. maximum reach extent, aperture, paw orientation, and maximum reach velocity) changed within individual sessions, a moving average was calculated by averaging kinematic data across a moving block of 10 trials. For averages within an experimental group, the last data point was carried forward to the end of the data set. This avoided sudden changes in the average caused by rats performing different numbers of trials within a session.

### Analysis of reach-to-grasp coordination

To monitor aperture and paw orientation as a function of the z-coordinate of the tip of the second digit ($z_{digit2}$, *Figures 8,9* and all *Figure 9—figure supplements 1–5*), the first reach of each trial was isolated. The three-dimensional trajectory of each digit tip for the initial reach was interpolated using piecewise cubic Hermite polynomials (*pchip* in MATLAB) so that the three-dimensional location of each digit was estimated for $z_{digit2} = -20.0, -19.9, -19.8 \ldots +14.9, +15.0$ mm from the pellet (positive numbers are past the pellet, negative numbers as the paw approaches the pellet). This allows us to average aperture and orientation as a function of paw advancement (assessed by $z_{digit2}$). Points at $z_{digit2}$ values missing from shorter reaches were excluded from the average. Note that trajectories in *Figure 9* could extend past the average $z_{digit2}$ endpoints in *Figure 6A* because of longer-than-average trajectories.

Two rats from the 'ChR2 During' group were excluded from the averaged aperture and paw orientation as a function of $z_{digit2}$ (*Figure 9B,E*). The majority of these rats' reaches during 'laser

sessions' were so short that there were not enough trials with full trajectories to produce a meaningful average (see *Figure 9—figure supplements 1,2* for analysis with all six rats).

To compare the evolution of aperture and paw orientation between retraining, laser, and occlusion sessions, we compared digit aperture and paw orientation at specific $z_{digit2}$ values. For all groups except 'ChR2 During' and 'ChR2 Between', we evaluated aperture and orientation at $z_{digit2}$ = 1 mm past the pellet ($z_{digit2}$ = +1 mm). Because rats frequently did not reach past the pellet when dopamine neurons were activated 'during reach', we analyzed aperture and paw orientation at $z_{digit2}$ = 7 mm before the pellet ($z_{digit2}$ = −7 mm) for this group.

## Abnormal involuntary movements (AIMs) testing

Rats underwent AIMs testing twice – one day before the first day of retraining and one day after the last day of occlusion sessions. One rat from the 'ChR2 During' group and one rat from the 'ChR2 Between' group were only tested after occlusion sessions, and therefore were not included in the analysis. Rats were attached to the patch cable and placed into a clear plexiglass cylinder (diameter = 21 cm). Two mirrors were placed behind the chamber so that the animal was visible in all positions in recordings. Once in the cylinder, animals underwent a series of 30 s stimulation epochs alternating with 30 s rest periods. Sessions always began with a rest period (baseline), and the order of laser power (estimated 5, 10, 15, 20, 25 mW at fiber tip) was randomly generated in Matlab. Stimulation was applied at 20 Hz at a 50% duty cycle. Stimulation sessions were video recorded at 60 frames-per-second (HC-V110, Panasonic).

AIMs videos were segmented into individual videos for each stimulation bout and assigned random codes so that scorers were blinded to the rat's virus (ChR2 or EYFP), laser power, and day of testing. Axial and limb AIMs were scored for both severity (amplitude scale) and duration (basic scale) (*Sebastianutto et al., 2016*). The amplitude and basic scores were multiplied to create a composite score for axial and limb AIMs. Global AIMs scores were the sum of the axial and limb composite scores. Rotational behavior was also analyzed by counting the number of full 360 degree rotations in the contralateral and ipsilateral directions during each 30 s video. Ipsilateral turns were subtracted from contralateral turns to identify a rotational bias.

To assess if AIMs were present during reaches, ten trials in which the rat failed to obtain the pellet ('pellet knocked off' or 'pellet remained' outcomes) were randomly selected for each rat from 'laser on' sessions 2 and 10 and 'occlusion' session 10 for 'ChR2 'during rats. Ten trials were also randomly selected for each EYFP rat from 'laser on' session 10. However, two EYFP rats only had 1 and 3 trials from 'laser on' session 10 with failed outcomes. Therefore, additional trials from 'laser on' sessions 9 (both rats) and 8 (one rat) were selected for these two rats. Once all videos were selected, they were assigned random codes so the scorer was blind to the group (ChR2 during vs EYFP) and session. Axial and limb AIMs during reaching movements were evaluated for severity (amplitude scale – see above). Global AIMs scores were calculated as the sum of the axial and limb amplitude scores for each rat.

## Immunohistochemistry

Rats were deeply anesthetized with isoflurane (5%) and transcardially perfused with cold saline followed by 4% paraformaldehyde. Brains were post-fixed for no more than 24 hr at 4°C, rinsed with saline, and moved through 20% and 30% sucrose solutions (in PBS) at 4°C. Sagittal sections (30 μm thickness) were taken around SNc and where the optical fiber was visible on a cryostat (Leica Microsystems). To verify localization of viral expression in dopamine neurons and optical fiber placement above SNc, we performed immunohistochemistry for TH and EYFP. Mounted sections were washed with PBS and incubated with Triton X-100 and PBS (PBS-Tx) for 15 min. Slides were then incubated in 5% normal donkey serum (NDS) for 1 hr before primary antibody incubation (mouse anti-GFP, 1:1500, Life Technologies; rabbit anti-TH, 1:2000, Millipore) overnight at room temperature with NDS and PBS-Tx. Sections were then washed with PBS-Tx and incubated with secondary antibodies (Alexa Fluor 488 donkey anti-mouse, 1:500, Life Technologies; Alexa Fluor 555 donkey anti-rabbit, 1:500, Fisher Scientific) for 2 hr at room temperature. After washing four times with PBS, sections were coverslipped with ProLong Diamond (Invitrogen), allowed to dry for 24 hr, and then imaged with an Axioskops 2 Plus microscope fitted with an Olympus DP72 camera.

Images were stitched together and TH- and EYFP-stained images were overlaid in Photoshop to verify localization of viral expression to dopamine neurons. Images were evaluated by two people blinded to the behavioral outcomes of the individual rats on (1) sufficient virus expression in SNc and striatal dopamine neurons and (2) location of fiber tip over SNc. Data from rats whose histology was evaluated as not meeting both of these criteria by both evaluators were removed from the analysis (n = 8 rats removed, *Figure 1—figure supplement 1*). To obtain coordinates of optical fiber tips, histology images were overlaid on sagittal brain atlas images of the approximate M-L coordinate (*Paxinos and Watson, 1998*) and A-P and D-V coordinates were ascertained.

## Statistics

To test whether fiber tip location differed between groups, one-way ANOVAs were performed separately for the A-P, M-L, and D-V dimensions (using MATLAB *anova1*). To determine if fiber tip location affected stimulation efficacy, we plotted the average maximum $z_{digit2}$ of 'laser on' session 10 as a function of fiber tip location for each rat and fit linear regressions (using MATLAB *corr*) to the plots. Each dimension (A-P, M-L, D-V) was analyzed separately for each group. To compare the duration of laser activation between 'during reach' and 'between reach' (always 5 s) groups (*Figure 1—figure supplement 1*), we used sign tests (using MATLAB *signtest*).

Linear mixed-effects models were used to evaluate the effects of laser on performance outcomes and reach kinematics over sessions. We implemented linear mixed-effects models (using R *lmer*) with random intercepts/effects for each rat (where effect of laser varied between rats) and main interaction effects of group, session number, and laser. Linear mixed-effects models included averages for all 22 sessions (retraining, laser on, and occlusion) for all rats. For normalized success rate and number of trials data, the inverse hyperbolic sine was taken before analysis in the linear mixed-effects model to deal with zeroes in the dataset. Post hoc contrast testing was performed on these linear mixed-effects models to make comparisons between specific sessions within groups (using R, 'contest1D'). Similar models were used to evaluate changes in aperture and paw orientation at specific $z_{digit2}$ coordinates in *Figure 9C F*. However, random effects were designated where the effect of session varied between rats.

To assess if reach kinematics with 'laser on' differed between successful and failed reaches, we implemented linear mixed-effects models (using R *lmer*) with random intercepts/effects for each rat and main interaction effects of group, session, laser, and outcome (*Figures 6–7*; *Figure 6—figure supplement 7*; *Figure 7—figure supplement 7*). Similar models were used to evaluate changes in aperture and paw orientation at specific $z_{digit2}$ coordinates in successful vs failed trials in *Figure 9—figure supplements 2,3*.

To assess if grasp kinematics (i.e. aperture and paw orientation) and success rate differed for extent-matched reaches under different stimulation conditions, we used paired *t*-tests (using MATLAB *ttest*) to compare 'laser on' to 'retraining' and 'laser on' to 'occlusion' at each final $z_{digit2}$ extent for each group (*Figure 6* and *Figure 7—figure supplement 8*). To assess the effect of 'during reach' laser stimulation on kinematics in ChR2 between rats (*Figure 10B* and *Figure 10—figure supplement 1*), we implemented a linear mixed-effects model with random intercepts/effects for each rat and a main effect of test session (occlusion day 10, laser during reach 1, or laser during reach 2).

To assess the effect of stimulation on reach kinematics in '5-off/5-on' sessions (*Figure 10*), we implemented a linear mixed-effects model with random intercepts/effects for each rat (where the effect of trial number within block varied between rats) and main interaction effects of laser and trial number within blocks.

To assess how AIMs changed from the first to second day of testing and under different laser powers, we implemented a linear mixed-effects model with random effects for each rat (where the effect of test number varied between rats) and main interaction effects of group, test number, and laser power. To assess if AIMs scores differed between ChR2 and EYFP rats on test day one we performed post hoc contrast testing (using R, 'contest1D') to compare ChR2 to EYFP at each laser power. To assess if AIMs scores during reaching differed between sessions we applied Kruskal-Wallis tests (using MATLAB *kruskalwallis*).

To assess differences between groups in within-session analyses, we applied Wilcoxon rank sum tests (using MATLAB *ranksum*) at each trial number, with a p cutoff of 0.01 for significance (e.g., *Figure 6D*). A priori power calculations were not performed as there were no similar prior studies of reaching kinematics under dopamine manipulations from which we could extrapolate possible

results. Sample size estimates were based on analysis of the ChR2 'During Reach' stimulation cohort, which revealed highly consistent results with n = 6 rats.

## Data files

Dependencies within the MATLAB code to generate the figures are available at GitHub (*Bova et al., 2020*).

Many source data files contain datasets which are used by multiple figures. For clarity, we have described the data within each source data file that are relevant to that particular figure.

## Acknowledgements

We thank Maya Hammoud, Kaitlyn Mulligan, Kat Dumoulin, and Kestutis Micke for help with behavioral training, behavioral scoring, and immunohistochemistry. We thank Dr. Roger Albin for reviewing early versions of this manuscript. This work was supported by the University of Michigan, an Nvidia GPU grant, the Brain Research Foundation, the University of Michigan Udall Center (NIH P50NS091856), NIH K08-NS072183, and NIH R56-NS109227.

## Additional information

### Funding

| Funder | Grant reference number | Author |
| --- | --- | --- |
| National Institutes of Health | K08-NS072183 | Daniel K Leventhal |
| Nvidia | GPU grant | Daniel K Leventhal |
| University of Michigan | | Daniel K Leventhal |
| Brain Research Foundation | BRF Seed Grant | Daniel K Leventhal |
| National Institutes of Health | P50NS091856 | Daniel K Leventhal |
| National Institutes of Health | R56-NS109227 | Daniel K Leventhal |

The funders had no role in study design, data collection and interpretation, or the decision to submit the work for publication.

### Author contributions

Alexandra Bova, Software, Formal analysis, Validation, Investigation, Visualization, Methodology, Writing - original draft, Writing - review and editing; Matt Gaidica, Amy Hurst, Investigation, Methodology; Yoshiko Iwai, Julia Hunter, Investigation; Daniel K Leventhal, Conceptualization, Resources, Data curation, Software, Formal analysis, Supervision, Funding acquisition, Validation, Investigation, Methodology, Writing - original draft, Writing - review and editing

### Author ORCIDs

Matt Gaidica (iD) https://orcid.org/0000-0002-0191-1899
Daniel K Leventhal (iD) https://orcid.org/0000-0001-8174-5933

### Ethics

Animal experimentation: This study was performed in strict accordance with the recommendations in the Guide for the Care and Use of Laboratory Animals of the National Institutes of Health. All of the animals were handled according to an approved institutional animal care and use committee (IACUC) protocol (#8407) of the University of Michigan. All surgery was performed under isoflurane anesthesia, and every effort was made to minimize suffering.

### Decision letter and Author response

Decision letter https://doi.org/10.7554/eLife.61591.sa1
Author response https://doi.org/10.7554/eLife.61591.sa2

## Additional files

### Supplementary files
• Transparent reporting form

### Data availability

Original video files and extracted deeplabcut coordinates in csv format are available publicly on fig-share (https://doi.org/10.6084/m9.figshare.c.5095484.v1).

The following dataset was generated:

| Author(s) | Year | Dataset title | Dataset URL | Database and Identifier |
|---|---|---|---|---|
| Leventhal D, Bova A | 2020 | Precisely-timed dopamine signals establish distinct kinematic representations of skilled movements | https://doi.org/10.6084/m9.figshare.c.5095484.v1 | figshare, 10.6084/m9.figshare.c.5095484.v1 |

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
