## [Decision Letter]

**Acceptance summary:**

This work provides important new insights into the role of dopamine signaling in acquisition of fine-skilled movements.

**Decision letter after peer review:**

Thank you for submitting your article "Precisely-timed dopamine signals establish distinct kinematic representations of skilled movements" for consideration by *eLife*. Your article has been reviewed by three peer reviewers, including Aryn H Gittis as the Reviewing Editor and Reviewer #1, and the evaluation has been overseen by Kate Wassum as the Senior Editor.

The reviewers have discussed the reviews with one another and the Reviewing Editor has drafted this decision to help you prepare a revised submission.

As the editors have judged that your manuscript is of interest, but as described below that revisions are required before it is published, we would like to draw your attention to changes in our revision policy that we have made in response to COVID-19 (https://elifesciences.org/articles/57162). First, because many researchers have temporarily lost access to the labs, we will give authors as much time as they need to submit revised manuscripts. We are also offering, if you choose, to post the manuscript to bioRxiv (if it is not already there) along with this decision letter and a formal designation that the manuscript is "in revision at *eLife*". Please let us know if you would like to pursue this option. (If your work is more suitable for medRxiv, you will need to post the preprint yourself, as the mechanisms for us to do so are still in development.)

Summary:

The study by Bova and colleagues describe how midbrain dopamine neurons affect skillful control of forearm movements in a reach-to-grasp pellet task. The study provides much needed data on how dopamine manipulations affect skilled, dexterous movements, as well as the valuable observation of a history dependence to dopamine stimulation effects, which could be important for interpreting past and future dopamine stimulation results in the literature. A particularly intriguing aspect of this study was the observation that SNc dopamine stimulation during reaching gradually shifts kinematics, but re-introducing the stimulation immediately recalls similar impaired movement changes even after the kinematics returned to baseline measurements (laser occluded). Overall, this is an exciting study that highlights how dopamine signaling influences immediate and long-term motor execution in a behavioral paradigm that quantifies complex movements. However, a number of concerns were raised during the review that should be addressed:

Essential revisions:

1) Behavioral quantification during the task:

a) How do reach kinematics differ between successful vs. failed reach attempts? Are the trajectories and coordination sequence for the failed trials (in EYFP rats) completely different from the aberrant movements observed with stimulation? Since the first reach success rate is around 50-60%, the stimulation might be reinforcing failed reaches on subsequent trials somehow.

b) It seems a bit arbitrary to focus on first reach success. Kinematic analysis should be shown with accuracy and rate. Do peak velocities and the trajectories change as the success rate drops? This could provide a more complete view of how stimulation is changing this complex behavior, i.e. transport phase, grasp, etc. The authors note that accuracy has a floor effect and could change for a multitude of reasons.

c) The methods allude to a trial-scoring system to address behavior heterogeneity but this data is not shown anywhere in the paper. How much heterogeneity was there, especially, how often did trial types 5-8 occur? Is the distribution affected by stimulation? Do rats begin to try other strategies than contralateral reaching as the stimulation sessions go on? And if ipsilateral reaches (trial type 8) happen with any frequency, are they affected by stimulation?

d) A concern was raised that undetected AIMs during the reach task might contribute to failed grasps. The authors state that there are few involuntary movements during reaches, but it was not clear what analyses were used to test for AIMs during the reach task.

2) Validation of optogenetic manipulations:

a) In Figure 1—figure supplement 2, viral expression looks fairly uniform but fiber placement in M-L and A-P dimensions is variable. This is problematic if there is a systemic difference between groups. This worry could be addressed by testing for significance in M-L and A-P placement between groups and summarizing those data in Figure 1—figure supplement 1. Is fiber placement related to stim efficacy within groups? The authors could test for correlation between M-L or A-P placement and stim effect for the "ChR2-during" and "Arch-during" groups.

b) One reviewer was concerned about the lack of measurements of increases or decreases in DA levels in the striatum during optogenetic manipulations. Ultimately, a discussion amongst the reviewers determined that due to precedence in the literature, such validation is not explicitly needed here. If the authors have any data to indicate the magnitude or duration of DA increases or decreases induced by optogenetic manipulations, or firing rate modulation of the TH neurons themselves, this would greatly strengthen the paper. However, the reviewers agreed that if such data could not be provided, it would not preclude publication of the study.

3) Methodological and statistical clarification.

a) Figure 7D: it is puzzling that while there is no overlap in the error bars, no significance is found, but for H, significance is occasionally found (likely a multiple comparisons effect). A correction for multiple comparisons seems necessary.

b) The authors write that during AIMs test 1, rats "appeared unaffected by dopamine neuron stimulation". This statement calls for a test of significance difference between AIMs scores in CHR2 test 1 and EYFP test 1.The data appear to be different, but there are no stats. If they are different, this would undercut the author's statement that reaches are unaffected by DA stimulation.

c). H was the LME model fit (ie in Figure 2, 3, 4, 6)? Was it between all baseline points and all points in the stimulation/control period per animal?

4) Interpretation:

a) The authors interpret the findings in Figures 10 and 11 as reflecting the establishment of "distinct reach-to-grasp representations" and behavioral context dependence. But couldn't they instead reflect a general sensitization to the particular exogenous DA stimulation being used? There has been no distinct representation formed for the AIMs test cylinder, yet on test 2 there are much larger effects on behavior. This doesn't seem the result of a context-dependent, distinct representation as much as an increase in general sensitivity to SNc-DA stimulation. I feel that conclusions about "representations" and "context" should be tempered a bit more.

b) The fact that SNc-DA inhibition reduces attempted reaches (Figure 4A) is "consistent with a role for midbrain dopamine in motivation to work for rewards," but then why doesn't stimulation increase attempted reaches?

---

## [Author Response]

Essential revisions:1) Behavioral quantification during the task:a) How do reach kinematics differ between successful vs. failed reach attempts? Are the trajectories and coordination sequence for the failed trials (in EYFP rats) completely different from the aberrant movements observed with stimulation? Since the first reach success rate is around 50-60%, the stimulation might be reinforcing failed reaches on subsequent trials somehow.

This is an interesting point, and in fact how we initially interpreted the data. After careful review, however, we decided that a pure “reinforcement” model cannot account for all of our observations. The reviewer’s comments prompted new analyses that helped clarify our thoughts.

We analyzed reach kinematics separately for successful and failed reaches. These new analyses are shown in Figure 6—figure supplement 7, Figure 7—figure supplement 7, and Figure 9—figure supplements 2 and 3. At baseline and in EYFP controls (also in Arch During and Between groups), there was a subtle, nonsignificant difference in reach extent between successful and failed reaches. Failed reaches tended to be slightly shorter, supporting the reviewer’s hypothesis that dopamine neuron stimulation might gradually reinforce “bad” kinematics.

While this analysis lends some support to a “reinforcement” model, we feel that it is more likely that dopamine has an intrinsic kinematic bias (though the two possibilities are not mutually exclusive). First, while there was a consistent small difference between successful and failed reaches, it did not reach significance in any analysis (other than “ChR2 during”, “laser on” sessions). We are therefore hesitant to build a model around this finding. More importantly, other observations are inconsistent with the reinforcement model. We stimulated/inhibited dopamine neurons on every trial. Thus, both long and short reaches should have been reinforced, effectively cancelling each other out. Furthermore, dopamine neuron inhibition caused reaches to extend further. By suppressing dopamine neuron activation on successful reaches, we would expect reaches to progressively shorten (presumably dopamine release would already be suppressed during failed short reaches). To definitively determine whether these effects are driven primarily by a reinforcement effect or an intrinsic kinematic bias, closed-loop experiments in which dopamine neurons are manipulated on only “short” or “long” reaches would be required. Recent advances like DeepLabCut-Live! for real-time motion tracking may make this possible.

We added text addressing these comments in the Discussion, beginning with “At baseline and in EYFP controls, failed reaches were slightly (but not significantly) shorter than successful reaches…”

b) It seems a bit arbitrary to focus on first reach success. Kinematic analysis should be shown with accuracy and rate. Do peak velocities and the trajectories change as the success rate drops? This could provide a more complete view of how stimulation is changing this complex behavior, i.e. transport phase, grasp, etc. The authors note that accuracy has a floor effect and could change for a multitude of reasons.

We appreciate the reviewer’s clarifications of this comment, which we address individually below:

1) Show overall accuracy for all attempts, not just first reach. For all conditions.

The requested plots have been added as supplements for Figures 2, 3, and 4. Overall, “any reach” success is higher than “first reach” success, but follows the same pattern.

2) How many attempts were present for each of the conditions?

These plots are included in the same supplements for Figures 2-4. Other than increased attempts in the “During Reach” ChR2 stimulation sessions, rats mostly made one attempt per trial.

3) Examine if reach amplitude explains the drop in accuracy. What is the accuracy for trials that appear to have normal reach transport? This comparison can be done for all.4. Are observed changes in grasp linked to reaching deficits. Are they compensatory for reaching deficits? Examine reaches that appear to be normal (if there) to see if grasp deficits are still there.

We grouped these together because they are addressed by the same new supplemental figures (Figure 6—figure supplement 8 and Figure 7—figure supplement 8). This is an interesting way of looking at the data that we had not considered, and thank the reviewer for suggesting it.

First, we excluded reaches where the digit did not extend to the pellet (i.e., z_digit2_ < 0). Then, we examined reaches of different extent in 1 mm bins past the pellet. Interestingly, for intermediate reach extension (i.e., ~1-7 mm past the pellet), “ChR2 during” rats were less successful in the “laser on” compared to the “occlusion” condition. As the reviewer suggested, we interpret this to mean that some factor (perhaps coordination of pronation/digit extension) influences success in addition to the change in reach extent/transport.

Nonetheless, there is still a large difference in reach extent between successful and failed reaches in the ChR2 during group (Figure 6—figure supplement 7). Changes in reach extent therefore also strongly contribute to the change in success rate.

Regarding the second question, there was a significant difference in paw orientation for extent-matched reaches between laser on and occlusion sessions for ChR2 during rats. This may explain some of the change in success rate at intermediate reach extents. We did not see any significant differences in aperture for extent-matched reaches.

We added sentences under “Dopamine manipulations induce progressive changes in reach-to-grasp kinematics” in the Results section to describe these findings: “The decrease in reach extent accounted for much of the drop in success rate (Figure 6—figure supplement 7), though reaches matched for reach extent were still less successful during “laser on” than “occlusion” sessions (Figure 6—figure supplement 8). This suggests that paw transport and grasp-related factors both contributed to reach-to-grasp failures.”

c) The methods allude to a trial-scoring system to address behavior heterogeneity but this data is not shown anywhere in the paper. How much heterogeneity was there, especially, how often did trial types 5-8 occur? Is the distribution affected by stimulation? Do rats begin to try other strategies than contralateral reaching as the stimulation sessions go on? And if ipsilateral reaches (trial type 8) happen with any frequency, are they affected by stimulation?

Figure 2—figure supplement 2, Figure 3—figure supplement 2, and Figure 4—figure supplement 3 show the requested data. The main differences that we saw with “during reach” stimulation were that the pellet was knocked off or remained on the pedestal more frequently. Although ipsilateral reaches were rare in the “ChR2 during” group in “laser on” sessions (only 2 of 6 rats had any ipsilateral reaches – 6 in one rat and 1 in the other rat), this achieved statistical significance because ipsilateral reaches almost never happened in “retraining” or “occlusion” sessions (2 of 6 rats in “occlusion” sessions – 1 trial each). There were also a couple of other significant effects – for example, there were a few more trials where no pellet was grabbed by the delivery arm in the “ChR2 between” group during “laser on” sessions. We agree that it would be interesting to see if stimulation also influences ipsilateral forelimb dexterity.

d) A concern was raised that undetected AIMs during the reach task might contribute to failed grasps. The authors state that there are few involuntary movements during reaches, but it was not clear what analyses were used to test for AIMs during the reach task.

We performed two additional analyses to address this question. First, to directly quantify the presence/absence of AIMs during reaches, we randomly sampled 10 reaches from “laser on” day 2, “laser on” day 10, and “occlusion” day 10 from each “ChR2 during” rat. We also randomly sampled 10 reaches from each “laser on” day 10 session in the EYFP group. These videos were scored by an investigator blinded to the session identity, rat identity, and experimental group (Ms. Hunter, who has been added as an author). Details of the scoring system have been added to the Materials and methods (paragraph beginning “To assess if AIMs were present during reaches…”), and are based on the system for scoring AIMs in the clear cylinder. These data are shown in the new Figure 11, panel A. Quantifying AIMs during reaches is difficult because the limb is already moving; discerning a superimposed abnormal movement is somewhat subjective. Nonetheless, the vast majority of reaches appeared truncated without superimposed adventitious movements. 3 rats accounted for all of the AIMs – 2 rats with one AIM each, and one rat with 5 (again, out of 10 sampled at random). Furthermore, superimposed AIMs were never observed in the early “laser on” sessions when success rate was already significantly decreased.

We also looked at whether the dyskinesia scores for individual rats predicted success rates during stimulation (Figure 11—figure supplement 1). In panel A, we looked at “ChR2 During” rats, and plotted the global AIMs score during the second AIMs test (at 20 mW at the fiber tip) against the change in success rate from retraining to “laser on” day 10. Several rats without dyskinesias still had significant drops in success rate, further arguing that dyskinesias are unlikely to account for most of the drop in success rate. In panel B, we looked at “ChR2 Between” rats, and plotted the global AIMs score during the second AIMs test (at 20 mW at the fiber tip) against the change in reach success rate from “occlusion” sessions to the first “laser on during” session. In this case, several rats with significant dyskinesias experienced no change in success rate. Thus, dyskinesias in the cylinder did not necessarily generalize to the reaching context.

Collectively these data suggest that dyskinesias may account for some reach failures in severely affected rats, but this is not a major failure mechanism. Examples of what we considered abnormal reach kinematics with and without superimposed dyskinesias are shown in Video 7.

2) Validation of optogenetic manipulations:a) In Figure 1—figure supplement 2, viral expression looks fairly uniform but fiber placement in M-L and A-P dimensions is variable. This is problematic if there is a systemic difference between groups. This worry could be addressed by testing for significance in M-L and A-P placement between groups and summarizing those data in Figure 1—figure supplement 1. Is fiber placement related to stim efficacy within groups? The authors could test for correlation between M-L or A-P placement and stim effect for the "ChR2-during" and "Arch-during" groups.

We added panels C and D to Figure 1—figure supplement 2 showing the distribution of optical fiber locations for each group along the A-P, M-L, and D-V axes. There were no significant differences across groups, and the locations largely overlapped. Therefore, fiber location is very unlikely to account for differences in outcome between groups. Furthermore, “during reach” stimulation following “between reach” stimulation verified that fibers were in the correct location to cause behavioral effects (Figure 10A-C and Figure 10—figure supplements 1 and 2).

To assess a possible relationship between fiber tip location and the magnitude of kinematic changes, we plotted reach extent (maximum *z_digit2_*) as a function of fiber tip coordinates along the A-P, M-L, and D-V axes for each group (the new Figure 6—figure supplement 1 and Figure 7—figure supplement 1). There was a suggestion that more posteromedially located fibers had stronger effects, suggesting that some of these effects could be mediated by activation of lateral VTA in addition to SNc. However, this effect was largely driven by one rat (the diamond in Figure 6—figure supplement 1 “ChR2 during” plots). We added a line under Dopamine manipulations induce progressive changes in reach-to-grasp kinematics describing this result: “This effect may have been stronger for posteromedially located fibers (Figure 6—figure supplement 1).”

b) One reviewer was concerned about the lack of measurements of increases or decreases in DA levels in the striatum during optogenetic manipulations. Ultimately, a discussion amongst the reviewers determined that due to precedence in the literature, such validation is not explicitly needed here. If the authors have any data to indicate the magnitude or duration of DA increases or decreases induced by optogenetic manipulations, or firing rate modulation of the TH neurons themselves, this would greatly strengthen the paper. However, the reviewers agreed that if such data could not be provided, it would not preclude publication of the study.

We do not have these data, but appreciate their relevance. This is a goal of ongoing work.

3) Methodological and statistical clarification.a) Figure 7D: it is puzzling that while there is no overlap in the error bars, no significance is found, but for H, significance is occasionally found (likely a multiple comparisons effect). A correction for multiple comparisons seems necessary.

The significance was marked in Figure 7I incorrectly (the significance threshold was incorrectly set to p < 0.05 instead of p < 0.01 as for panels E and G). This error has been corrected and all significance bars for within session analyses now reflect p < 0.01.

b) The authors write that during AIMs test 1, rats "appeared unaffected by dopamine neuron stimulation". This statement calls for a test of significance difference between AIMs scores in CHR2 test 1 and EYFP test 1.The data appear to be different, but there are no stats. If they are different, this would undercut the author's statement that reaches are unaffected by DA stimulation.

We performed post hoc contrast tests to directly compare ChR2 test 1 and EYFP test 1 at every power. There were no significant differences on test day 1 between ChR2 and EYFP rats at any power for global, limb, or axial AIMs.

c). H was the LME model fit (ie in Figure 2, 3, 4, 6)? Was it between all baseline points and all points in the stimulation/control period per animal?

The LME model was fit using averaged data from “retraining”, “laser on” and “occluded” sessions for each rat in each group. Comparisons were made between “laser on,” “retraining,” and “occluded” sessions.

4) Interpretation:a) The authors interpret the findings in Figures 10 and 11 as reflecting the establishment of "distinct reach-to-grasp representations" and behavioral context dependence. But couldn't they instead reflect a general sensitization to the particular exogenous DA stimulation being used? There has been no distinct representation formed for the AIMs test cylinder, yet on test 2 there are much larger effects on behavior. This doesn't seem the result of a context-dependent, distinct representation as much as an increase in general sensitivity to SNc-DA stimulation. I feel that conclusions about "representations" and "context" should be tempered a bit more.

We agree with the reviewer that conceptualizing the changes as “sensitized” reaching instead of distinct kinematic representations makes sense, especially since the reaches were qualitatively similar (but compressed or expanded) between stimulated, inhibited, and control rats. In fact, we rewrote the relevant paragraphs (beginning with “Kinematic changes developed gradually, but once established they depended on the dopamine status of the current trial (Figure 10).”) to frame the kinematic changes as a sensitization effect akin to sensitization to drugs of abuse. We believe this is an important idea that provides direction for future studies investigating the mechanism(s) of these effects. We thank the reviewer for this insight.

However, this sensitization effect is specific based on two key observations. If there were a general sensitization effect, then “during reach” stimulation should immediately cause markedly abnormal reaches after “between reach” stimulation. To test this possibility, we examined these sessions in more detail (new Figures 10A-C and Figure 10—figure supplements 1 and 2). “During reach” stimulation did not immediately cause abnormal reach kinematics in rats that had previously received “between reach” stimulation. Instead, changes in reach kinematics progressed gradually – in fact, they followed a pattern highly similar to initial stimulation in the “During ChR2” group.

The context-dependence of these effects is illustrated by the comparison between dyskinesias in the cylinder and during reaching. As shown in the new Figure 11A (and Figure 11—figure supplement 1, see response to point 1d), rats had rare limb dyskinesias during reaching in the session immediately preceding the second AIMs test. It is difficult to quantitatively compare AIMs severity during reaching and in the cylinder because the paw is already in motion during reaching. However, compare the severe axial and limb dyskinesias in the cylinder in Video 6 to the limited, relatively mild dyskinesias during reaching in the same rat at the same stimulation amplitude (Videos 6 and 7). In summary, while dyskinesias may occur in either the cylinder or reaching chamber at the same stimulation settings, they were more common and more severe in the cylinder. They are therefore context-dependent.

b) The fact that SNc-DA inhibition reduces attempted reaches (Figure 4A) is "consistent with a role for midbrain dopamine in motivation to work for rewards," but then why doesn't stimulation increase attempted reaches?

There was a non-significant increase in the number of trials performed for both the “during reach” and “between reach” ChR2 groups. However, well-trained rats initiate new trials quickly, even in the absence of stimulation. They are likely already performing a near-maximum number of reaches at baseline. This “ceiling effect” is the most likely explanation for why rats did not perform additional reaches. A sentence was added to note this possibility (“It is not clear why dopamine neuron stimulation did not have the opposite effect…”).